# DISTRIBUTIONAL PERTURBATION FOR EFFICIENT EXPLORATION IN DISTRIBUTIONAL REINFORCEMENT LEARNING

## ABSTRACT

Distributional reinforcement learning aims to learn distribution of return under stochastic environments. Since the learned distribution of return contains rich information about the stochasticity of the environment, previous studies have relied on descriptive statistics, such as standard deviation, for optimism in the face of uncertainty. However, using the uncertainty from an empirical distribution can hinder convergence and performance when exploring with the certain criterion that has an one-sided tendency on risk in these methods. In this paper, we propose a novel distributional reinforcement learning that explores by randomizing risk criterion to reach a risk-neutral optimal policy. First, we provide a perturbed distributional Bellman optimality operator by distorting the risk measure in action selection. Second, we prove the convergence and optimality of the proposed method by using the weaker contraction property. Our theoretical results support that the proposed method does not fall into biased exploration and is guaranteed to converge to an optimal return distribution. Finally, we empirically show that our method outperforms other existing distribution-based algorithms in various environments including Atari games.

## 1 INTRODUCTION

Distributional reinforcement learning (DRL) learns the stochasticity of returns in the reinforcement learning environments and has shown remarkable performance in several benchmark tasks. Its model generates the approximated distribution of returns, where the mean value implies the traditional Q-value (Bellemare et al., 2017; Dabney et al., 2018b; Choi et al., 2019). Learning procedure with stochasticity through return distribution is represented by *parametric (epistemic) uncertainty*, which is due to insufficient or inaccurate data, and *intrinsic (aleatoric) uncertainty*, which is inherently possessed randomness in the environment. (Chow et al., 2015; Dabney et al., 2018a) The learned stochasticity gives rise to the notion of risk-sensitivity, and some distributional reinforcement learning algorithms distort the learned distribution to create a risk-averse or risk-seeking policy.

Another way to employ the uncertainty is to design an efficient exploration method which is essential to find an optimal behavior with a few number of trials. *Optimism in the face of uncertainty* (OFU) is one of the fundamental exploration principles that employs parametric uncertainty to promote exploring less understood behaviors and to construct confidence set. Most OFU algorithms select an action with the highest upper-confidence bound (UCB) of uncertainty which can be considered as the optimistic decision at the moment (Chen et al., 2017; Ciosek et al., 2019). In deep RL, several OFU studies often model the uncertainty explicitly through the Bayesian posterior, which is estimated by using neural networks. However, learning the representation of high-dimensional state-action space and Bellman update simultaneously leads to unstable propagation(Yang et al., 2021).

On the other hand, DRL can provide more statistical information during control such as mode, median, or variance by addressing full characteristics of the return distribution. Despite the richness of information for return distribution, only a few DRL methods have tried to employ the benefits of distributional perspective for exploration (Tang & Agrawal, 2018; Mavrin et al., 2019; Clements et al., 2019). To estimate the uncertainty from distributional outputs, descriptive statistics that is composed of a mixture of intrinsic and parametric uncertainty can be utilized. Unfortunately, sepa-

rating these two types of uncertainty during learning is not a trivial task. Mavrin et al. (2019) propose a distribution-based OFU exploration that schedules a decaying bonus rate to suppress the effect of intrinsic uncertainty, which unintentionally induces a risk-seeking policy. Although OFU based approaches try to reduce parametric uncertainty by revisiting the state with high uncertainty, there exists the side effect that the criteria unfortunately forces the agent to chase the intrinsic uncertainty (risk) simultaneously during updates. Relying on a specific criteria causes an one-sided tendency on risk and makes an agent consistently select certain actions during exploration as degrading performance. We call this phenomenon as *stuckness*.

In this paper, we introduce *perturbed quantile regression* (PQR) which perturbs the criterion on uncertainty by randomizing the risk criterion in action selection. First, the distributional perturbation on return distribution is to re-evaluate the estimate of return by distorting the learned distribution with perturbation weight. Unlike the typical worst-case approach in risk-sensitive settings or OFU based approaches, we instead randomly sample a risk measure from an ambiguity set, which represents that the risk setting is ambiguous when the characteristics of a given environment are unknown. We empirically demonstrate on the stochastic variant of N-Chain environment (Osband et al., 2016) that a randomized scheme is more effective than OFU to alleviate the sub-optimality problem of tendency to obtain risk-seeking policies. Second, any risk-measure with some time-varying perturbation constraint allows us to average over all possible risk-sensitive behaviors to achieve the optimal policy. We provide a sufficient condition for the convergence of return distribution in the weaker contraction property. Our method covers the full elements of the reinforcement learning with distributional perspective from the novel exploration method with distributional perturbation to the theoretically guaranteed convergence of return distribution, which has the same fixed-point with the standard Bellman optimality operator. The proposed algorithm is based on QR-DQN, which is a baseline of DRL architecture to learn a return distribution, and we show that PQR outperforms QR-DQN and other DRL baselines in several benchmark environments, LunarLander-v2 and Atari games.

## 2 BACKGROUNDS & RELATED WORKS

### 2.1 DISTRIBUTIONAL REINFORCEMENT LEARNING

We consider a Markov decision process (MDP) which is defined as a tuple $(\mathcal{S}, \mathcal{A}, P, R, \gamma)$ where $\mathcal{S}$ is a finite state space, $\mathcal{A}$ is a finite action space, $P : \mathcal{S} \times \mathcal{A} \times \mathcal{S} \to [0, 1]$ is the transition probability, $R$ is the random variable of rewards in $[-R_{\max}, R_{\max}]$, and $\gamma \in [0, 1)$ is the discount factor. We define a stochastic policy $\pi(\cdot|s)$ which is a conditional distribution over $\mathcal{A}$ given state $s$. For a fixed policy $\pi$, we denote $Z^\pi(s, a)$ as a random variable of return distribution of state-action pair $(s, a)$ following the policy $\pi$. We attain $Z^\pi(s, a) = \sum_{t=0}^\infty \gamma^t R(S_t, A_t)$, where $S_{t+1} \sim P(\cdot|S_t, A_t)$, $A_t \sim \pi(\cdot|S_t)$ and $S_0 = s$, $A_0 = a$. Then, we define an action-value function as $Q^\pi(s, a) = \mathbb{E}[Z^\pi(s, a)]$ in $[-V_{\max}, V_{\max}]$ where $V_{\max} = R_{\max}/(1-\gamma)$. For regularity, we further notice that the space of action-value distributions $\mathcal{Z}$ has the first moment bounded by $V_{\max}$:

$$\mathcal{Z} = \left\{ Z : \mathcal{S} \times \mathcal{A} \to \mathscr{P}(\mathbb{R}) \big| \; \mathbb{E}[|Z(s, a)|] \leq V_{\max}, \forall (s, a) \right\}.$$

In distributional RL, the return distribution for the fixed $\pi$ can be computed via dynamic programming with the distributional Bellman operator defined as (Bellemare et al., 2017),

$$\mathcal{T}^\pi Z(s, a) \overset{D}{=} R(s, a) + \gamma Z(S', A'), \quad S' \sim P(\cdot|s, a), \; A' \sim \pi(\cdot|S')$$

where $\overset{D}{=}$ denotes that both random variables share the same probability distribution. We can compute the optimal return distribution by using the distributional Bellman optimality operator defined as,

$$\mathcal{T} Z(s, a) \overset{D}{=} R(s, a) + \gamma Z(S', a^*), \quad S' \sim P(\cdot|s, a), \; a^* = \arg\max_{a'} \mathbb{E}_Z[Z(S', a')].$$

Bellemare et al. (2017) have shown that $\mathcal{T}^\pi$ is a contraction in a maximal form of the Wasserstein metric but $\mathcal{T}$ is not a contraction in any metric. Combining with the expectation operator, $\mathbb{E}\mathcal{T}$ is a contraction so that we can guarantee that the expectation of $Z$ converges to the optimal state-action value, while the convergence of a return distribution itself is not guaranteed.

## 2.2 EXPLORATION ON DISTRIBUTIONAL REINFORCEMENT LEARNING

In this section, we will briefly describe the main practical algorithm for DRL with a deep neural network, and explain the derived exploration method. To combine with deep RL, a parametric distribution $Z_\theta$ is used to learn a return distribution by using $\mathcal{T}$. Dabney et al. (2018b) have employed a quantile regression to approximate the full distribution by letting $Z_\theta(s, a) = \frac{1}{N} \sum_{i=1}^{N} \delta_{\theta_i(s,a)}$ where the parameter $\theta$ represents the locations of a mixture of $N$ Dirac delta functions. Each $\theta_i$ represents the value where the cumulative probability is $\tau_i = \frac{i}{N}$. Then, by using the quantile representation with the distributional Bellman optimality operator, the problem can be formulated as a minimization problem as,

$$\theta = \arg\min_{\theta'} D\left(Z_{\theta'}(s_t, a_t), \mathcal{T} Z_{\theta^-}(s_t, a_t)\right) := \sum_{i=1}^{N} \sum_{j=1}^{N} \frac{1}{N} [\rho_{\hat{\tau}_i}^\kappa (r_t + \gamma \theta_j^-(s_{t+1}, a') - \theta_i'(s_t, a_t))]$$

where $(s_t, a_t, r_t, s_{t+1})$ is a given transition pair, $\hat{\tau}_i = \frac{\tau_{i-1} + \tau_i}{2}$, $a' := \arg\max_{a'} \mathbb{E}_Z[Z_\theta(s_{t+1}, a')]$, $\rho_{\hat{\tau}_i}^\kappa(x) := |\hat{\tau}_i - \delta_{\{x<0\}}| \mathcal{L}_\kappa(x)$, and $\mathcal{L}_\kappa(x) := x^2/2$ for $|x| \le \kappa$ and $\mathcal{L}_\kappa(x) := \kappa(|x| - \frac{1}{2}\kappa)$, otherwise. Based on the quantile regression, Dabney et al. (2018b) have proposed a quantile regression deep Q network (QR-DQN) that shows better empirical performance than the categorical approach (Bellemare et al., 2017) since the quantile regression does not restrict the bounds for return. As deep RL typically did, QR-DQN adjusts $\epsilon$-greedy schedule, which selects the greedy action with probability $1 - \epsilon$ and otherwise selects random available actions uniformly. The majority of QR-DQN variants (Dabney et al., 2018a; Yang et al., 2019) rely on the same exploration method. However, such approaches do not put aside inferior actions from the selection list and thus suffers from a loss (Osband et al., 2019). Hence, selecting a statistically plausible action is crucial for efficient exploration.

In recent studies, Mavrin et al. (2019) modifies the criterion of selecting an action for efficient exploration in the face of uncertainty. Using left truncated variance as a bonus term to estimate optimistic way and decaying ratio $c_t$ to suppress the intrinsic uncertainty, DLTV was proposed as an uncertainty-based exploration in DRL without using $\epsilon$-greedy exploration. At timestep $t$, the action selection of DLTV can be described as:

$$a^* = \arg\max_{a'} \left( \mathbb{E}_P[Z(s', a')] + c_t \sqrt{\sigma_+^2(s', a')} \right), \quad c_t = c\sqrt{\frac{\log t}{t}}, \quad \sigma_+^2 = \frac{1}{2N} \sum_{i=\frac{N}{2}}^{N} (\theta_{\frac{N}{2}} - \theta_i)^2,$$

where $\theta_i$'s are the values of quantile level $\tau_i$. DLTV shows that a constant schedule degrades the performance significantly compared to a decaying schedule.

## 2.3 RISK IN DISTRIBUTIONAL REINFORCEMENT LEARNING

Instead of an expected value, risk-sensitive RL tries to maximize a risk measure such as Mean-Variance (Zhang et al., 2020), Value-at-Risk (VaR) (Chow et al., 2017), or Conditional Value-at-Risk (CVaR) (Rockafellar et al., 2000; Rigter et al., 2021), which result in different classes of optimal policy. Especially, Dabney et al. (2018a) interprets risk measures as the expected utility function of the return, i.e., $\mathbb{E}_Z[U(Z(s, a))]$. Under this interpretation, risk-sensitive RL can be formulated as the maximization problem with various types of utility functions. If the utility function $U$ is linear, the policy obtained under such risk measure is called *risk-neutral*. If $U$ is concave or convex, the resulting policy is termed as *risk-averse* or *risk-seeking*, respectively. In general, a *distortion risk measure* is a generalized expression of risk measure generated from the distortion function.

**Definition 1.** *Let $h : [0, 1] \to [0, 1]$ be a **distortion function** such that $h(0) = 0, h(1) = 1$ and non-decreasing. Given a probability space $(\Omega, \mathcal{F}, \mathbb{P})$ and a random variable $Z : \Omega \to \mathbb{R}$, a **distortion risk measure** $\rho_h$ corresponding to a distortion function $h$ is defined by:*

$$\rho_h(Z) := \mathbb{E}^{h(\mathbb{P})}[Z] = \int_{-\infty}^{\infty} z \frac{\partial}{\partial z} (h \circ F_Z)(z) dz,$$

*where $F_Z$ is the cumulative distribution function of $Z$.*

In fact, non-decreasing property of $h$ makes it possible to distort the distribution of $Z$ while satisfying the fundamental property of CDF. Note that the concavity and the convexity of distortion function

Figure 1: Illustration of the N-Chain environment starting from state $s_2$. To emphasize the stochasticity, the reward of state $s_4$ was set as a mixture model composed of two Gaussian distributions. Blue arrows indicate the risk-neutral optimal policy in this MDPs.

also imply risk-averse or risk-seeking behavior, respectively. Dhaene et al. (2012) showed that any distorted expectation can be expressed as weighted averages of quantiles. In other words, generating a distortion risk measure is equivalent to choosing a reweighting distribution.

Fortunately, distributional RL has a suitable configuration to apply those uncertainty-based approaches that could naturally expand the class of policies. Chow et al. (2015) and Stanko & Macek (2019) considered risk-sensitive RL with a CVaR objective, where risk is related to robust decision making. Dabney et al. (2018a) expanded the class of policies on arbitrary distortion risk measures and investigated the effects of a distinct distortion risk measures by changing the sampling distribution for quantile targets $\tau$. Unlike the usual risk-sensitive RL, DLTV applied the risk measure only on action selection, while it keeps the standard objective to obtain a risk-neutral optimal policy. Our analysis shows that risk-based exploration can utilize risk measures in two different ways: (1) selecting action and (2) evaluating the value function by using distorted (perturbed) expectation.

## 3 PERTURBATION IN DISTRIBUTIONAL REINFORCEMENT LEARNING

### 3.1 MOTIVATION

Distribution-based OFU exploration (Moerland et al., 2018; Keramati et al., 2020) was proposed to give a bonus for the uncertainty that can be extracted from the distribution. However, we found that keeping optimism on uncertainty tends to select sub-optimal behaviors over a long exploration. For example, suppose we choose a criterion based on mean-standard deviation with coefficient $c_t$. Consider two actions $a_1, a_2$ with mean $\mu_1, \mu_2$ and variance $\sigma_1, \sigma_2$ respectively, under the following conditions: $\mu_1 \geq \mu_2, \sigma_1 \leq \sigma_2,$ and $\mu_1 + c_t\sigma_1 \leq \mu_2 + c_t\sigma_2$. Then, the agent prefers to select $a_2$ based on OFU. To change the decision towards the true optimal action $a_1$, the following steps need to be spent:

$$\eta = \min\left\{t' > t : c_{t'} \leq \frac{\mu_1 - \mu_2}{\sigma_2 - \sigma_1}\right\} - t.$$

Hence, if there is a bias in the criterion itself, such stuckness often occurs and degrades the performance since the agent does not have experience with the optimal policy during that period.

To demonstrate such shortcomings of OFU exploration in distributional RL framework, we build a representative environment that is easy to interpret intuitively among the cases in which intrinsic uncertainty exists. We experiment on the stochastic variant of N-Chain environment used in Osband et al. (2016) as a toy experiment. A schematic diagram of the N-Chain environment is shown in Figure 1. The reward is only given in the leftmost and rightmost states and the game terminates when one of the reward states is reached. We set the leftmost reward as $\mathcal{N}(10, 0.1^2)$ and the rightmost reward as $\frac{1}{2}\mathcal{N}(5, 0.1^2) + \frac{1}{2}\mathcal{N}(13, 0.1^2)$ which has a lower mean as 9 but higher variance. The agent always starts from the middle state $s_2$ and should move toward the leftmost state $s_0$ to achieve the greatest expected return. For each state, the agent can take one of six available actions: left, right, and 4 no-op actions. The optimal policy with respect to mean is to move left twice from the start. Despite the simple configuration, the possibility to obtain a higher reward in the suboptimal state than the optimal state makes the agent difficult which policy is optimal until it experiences enough to detect the characteristics of each distribution. Thus, the goal of our toy experiment is to evaluate how quickly each algorithm could find a risk-neutral optimal policy.

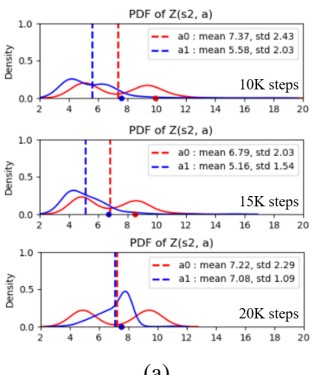 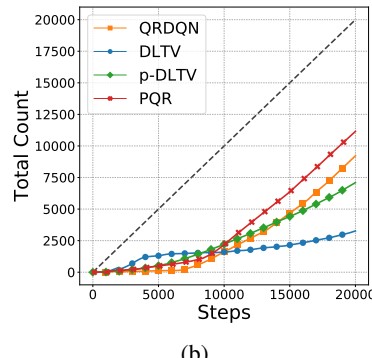

(a)        (b)

Figure 2: (a) Empirical return distribution of DLTV during training in N-Chain environment. The dashed lines denote the exact mean, and the dots on the x-axis denote the perturbed mean of each action. No-op actions are not shown for visibility. (b) Total count of performing true optimal action. The oracle (dashed line) is to perform the true optimal action from start to end.

Rather than applying a decaying schedule to suppress intrinsic uncertainty, we naively modify the optimism into a randomized risk criterion and named the perturbed variant as p-DLTV. We compare QR-DQN and DLTV with our randomized variants of two algorithms PQR and p-DLTV to examine the effect of randomized risk criteria. In short, p-DLTV is a straightforward modification of DLTV where randomness is given to the coefficient $c_t$ through normal distribution. We will describe our main algorithm, PQR, in detail in Section 3.4.

In Figure 2(a), DLTV fails to estimate the true optimal return distribution of action $a_1$. Due to the erroneous estimation, the agent takes longer to recognize its error. Hence, the deterministic selection based on a fixed criteria could mislead toward exploitation rather than exploration. This indicates that DLTV may not gather experiences well in a complex environment that requires deep exploration. In Figure 2(b), we plot the point when the optimal policy was actually performed for each algorithm to show stuckness. Since the optimal policy consists of only the same index $a_1$, we plot the total count of performing the optimal action with 10 different seeds. The slope indicates the ratio of performing the optimal policy on average. Hence, the interval with a slope of 1 implies that the optimal policy was performed every time. From the slope of each line, it is observed that DLTV selects the suboptimal action even if the optimal policy was initially performed. Although the mean return of $a_1$ (move left) is estimated to be superior, the agent only selects $a_2$ (move right) during training due to its consistent optimism on uncertainty. Even if DLTV has spent enough number of time steps to choose the true optimal policy, the remaining procedure is already close to greedy selection as it starts from a decreased coefficient.

Surprisingly, p-DLTV alleviates the stuckness early by randomly choosing a plausible action. We also propose a theoretically guaranteed algorithm to be converged, PQR which shows a much steeper line by quickly obtaining the optimal policy. In Section 3.2, we derive the first theoretical sufficient condition for the convergence of exploration method for DRL, which implies that PQR has the same unique fixed point as the standard distributional Bellman equation.

## 3.2 PERTURBED DISTRIBUTIONAL BELLMAN OPTIMALITY OPERATOR

To choose statistically plausible actions which may be maximal for certain criteria, we are interested in generating a distortion risk measure involved in a pre-defined constraint set called an *ambiguity set*. The ambiguity set, originated from distributionally robust optimization (DRO) literature, is a family of distribution characterized by a certain statistical distance such as $\phi$-*divergence* or *Wasserstein distance* (Esfahani & Kuhn, 2018; Shapiro et al., 2021). In this paper, we will examine the ambiguity set defined by the discrepancy between distortion risk measure and expectation. We say the sampled reweighting distribution $\xi$ as *(distributional) perturbation* and define it as follows:

**Definition 2.** *(Perturbation, Perturbation Gap, and Ambiguity Set) Given a probability space $(\Omega, \mathcal{F}, \mathbb{P})$, let $X : \Omega \to \mathbb{R}$ be a random variable and $\Xi = \left\{ \xi : \xi(w) \geq 0, \ \int_{w \in \Omega} \xi(w)\mathbb{P}(w)dw = 1 \right\}$*

*be a set of probability density functions. For a given constraint set $\mathcal{U} \subset \Xi$, we say $\xi \in \mathcal{U}$ as a* **(distributional) perturbation** *from $\mathcal{U}$ and denote the $\xi-$weighted expectation of $X$ as follows:*

$$\mathbb{E}_\xi[X] \coloneqq \int_{w \in \Omega} X(w)\xi(w)\mathbb{P}(w)dw,$$

*which can be interpreted as the expectation of $X$ under perturbed probability distribution $\xi\mathbb{P}$. We further define $d(X; \xi) = |\mathbb{E}[X] - \mathbb{E}_\xi[X]|$ as* **perturbation gap** *of $X$ with respect to $\xi$. Then, for a given constant $\Delta \geq 0$, we define the* **ambiguity set** *with the bound $\Delta$ as*

$$\mathcal{U}_\Delta(X) = \Big\{ \xi \in \Xi : d(X; \xi) \leq \Delta \Big\}.$$

For brevity, we omit the input $w$ from a random variable unless confusing. Since $\xi$ is a probability density function, $\mathbb{E}_\xi[X]$ is an induced risk measure with respect to a reference measure $\mathbb{P}$. Intuitively, $\xi(w)$ can be viewed as a distortion to generate a different probability measure and allow to vary the risk tendency. The aspect of using distortion risk measures looks similar to IQN (Dabney et al., 2018a). However, instead of changing the sampling distribution of quantile level $\tau$ implicitly, we reweight each quantile from the ambiguity set. This allows us to control the maximum allowable distortion with bound $\Delta$, whereas in IQN the risk measure does not change throughout learning. In Section 3.4, we will suggest a practical method to construct the ambiguity set.

Now, we characterize *perturbed distributional Bellman optimality operator* $\mathcal{T}_\xi$ for a fixed perturbation $\xi \in \mathcal{U}_\Delta(Z)$ written as below:

$$\mathcal{T}_\xi Z(s,a) \stackrel{D}{=} R(s,a) + \gamma Z(S', a^*(\xi)), \quad S' \sim P(\cdot|s,a), \; a^*(\xi) = \arg\max_{a'} \mathbb{E}_{\xi,P}[Z(s',a')].$$

Notice that $\xi \equiv 1$ corresponds to a base expectation, i.e., $\mathbb{E}_{\xi,P} = \mathbb{E}_P$, which recovers the standard distributional Bellman optimality operator $\mathcal{T}$. To understand the progress of PDBOO intuitively, we add the pipeline figure in Appendix A.4.

Specifically, updating via $\mathcal{T}_\xi$ can be described in two steps. 1) Select the superior action in terms of an arbitrary distortion risk measure instead of mean. We specify the distortion risk measure by sampling the perturbation $\xi$ from the ambiguity set. 2) Compute the return distribution through dynamic programming which follows the standard Bellman optimality equation. In risk-sensitive DRL or distributionally robust RL, the Bellman optimality equation is reformulated for a pre-defined risk measure (Chow et al., 2015; Smirnova et al., 2019; Yang, 2020). In this sense, PDBOO has a significant distinction in that it performs dynamic programming that adheres to the risk-neutral optimal policy while randomizing the risk criterion at every step.

If we consider the time-varying bound of ambiguity set, scheduling $\Delta_t$ is a key ingredient to determine whether PDBOO will efficiently explore or converge. The following lemma plays an important role to guarantee the convergence of PDBOO.

**Lemma 3.** *If $\xi_t$ converges to 1 uniformly on $\Omega$, then $\mathbb{E}\mathcal{T}_{\xi_t}$ also converges to $\mathbb{E}\mathcal{T}$ uniformly on $\mathcal{Z}$ for all $s \in \mathcal{S}$ and $a \in \mathcal{A}$.*

Intuitively, if an agent continues to sample the distortion risk measure from a fixed ambiguity set with a constant $\Delta$, there is a possibility of selecting sub-optimal actions after sufficient exploration, which may not guarantee eventual convergence. Hence, it will be crucial to schedule a constraint of ambiguity set properly at each action selection to guarantee convergence.

Based on the quantile model $Z_\theta$, our algorithm can be summarized into two parts. First, we aim to minimize the expected discrepancy between $Z_\theta$ and $\mathcal{T}_\xi Z_{\theta^-}$ where $\xi$ is sampled from ambiguity set $\mathcal{U}_\Delta$. To clarify notation, we write $\mathbb{E}_\xi[\cdot]$ as a $\xi-$weighted expectation and $\mathbb{E}_{\xi \sim \mathscr{P}(\mathcal{U}_\Delta)}[\cdot]$ as an expectation with respect to $\xi$ which is sampled from $\mathcal{U}_\Delta$. Then, our goal is to minimize the perturbed distributional Bellman objective with sampling procedure $\mathscr{P}$:

$$\min_{\theta'} \mathbb{E}_{\xi_t \sim \mathscr{P}(\mathcal{U}_{\Delta_t})}[D(Z_{\theta'}(s,a), \mathcal{T}_{\xi_t} Z_{\theta^-}(s,a))] \tag{1}$$

where we use the Huber quantile loss as a discrepancy on $Z_{\theta'}$ and $\mathcal{T}_\xi Z_{\theta^-}$ at timestep $t$. It is different from DRO which performs the worst-case optimization by using a minimax objective. By using expectation instead of max operator, we investigate risk-neutral exploration that can avoid overly

pessimistic policies. Second, considering a sequence $\xi_t$ which converges uniformly to 1 so that $\mathcal{T}_{\xi_t}$ converges uniformly to original $\mathcal{T}$, we derive a sufficient condition of $\Delta_t$ that the expectation of iterated operator $\mathbb{E}\mathcal{T}_{\xi_{n:1}} = \mathbb{E}\mathcal{T}_{\xi_n}\mathcal{T}_{\xi_{n-1}}\cdots\mathcal{T}_{\xi_1}$ has a unique fixed point with the same solution as the standard solution.

### 3.3 CONVERGENCE OF THE PERTURBED DISTRIBUTIONAL BELLMAN OPTIMALITY OPERATOR

In this section, we will provide the theoretical result of PDBOO about its convergence through $\mathbb{E}[Z^{(n)}]$ where the iteration procedure is denoted as $Z^{(n+1)} := \mathcal{T}_{\xi_{n+1}}Z^{(n)}$ and $Z^{(0)} = Z$ for each timestep $n > 0$.

**Theorem 4.** *For a sequence of bound $\Delta_n$, let $\bar{\mathcal{U}}_{\Delta_n}(Z^{(n-1)}) := \bigcap_{s,a}\mathcal{U}_{\Delta_n}\left(Z^{(n-1)}(s,a)\right)$. If we sample $\xi_n$ from $\bar{\mathcal{U}}_{\Delta_n}(Z^{(n-1)})$ for every iteration and $\sum_{n=1}^{\infty}\Delta_n < \infty$ holds, then, the expectation of iterated operator $\mathcal{T}_{\xi_{n:1}}$ has a fixed point $\mathbb{E}[Z^*]$. Moreover, the following bound holds,*

$$\sup_{s,a}\left|\mathbb{E}[Z^{(n)}(s,a)] - \mathbb{E}[Z^*(s,a)]\right| \leq \sum_{k=n}^{\infty}\left(2\gamma^{k-1}V_{max} + 2\sum_{i=1}^{k}\gamma^i(\Delta_{k+2-i} + \Delta_{k+1-i})\right).$$

Theorem 4 states that $\mathcal{T}_{\xi_{n:1}}$ converges while it does not satisfy the $\gamma$-contraction property. Note that the fixed point $\mathbb{E}[Z^*]$ is not yet guaranteed to be unique for any $Z \in \mathcal{Z}$. Fortunately, we could show that $\mathbb{E}[Z^*]$ is, in fact, the solution of the standard Bellman optimality equation which is already well known to have a unique solution.

**Theorem 5.** *If $\{\Delta_n\}$ follows the assumption in Theorem 4, then $\mathbb{E}[Z^*]$ is the unique solution of Bellman optimality equation.*

From two theoretical results, PDBOO is guaranteed to have the same unique fixed point as the standard Bellman operator under weaker contraction property. Unlike other distribution-based or risk-sensitive approaches, PDBOO is a novel operator having compatibility for obtaining a risk-neutral optimal policy. Also, it is also observed that this randomized method is more efficient than the OFU-based exploration by quickly alleviating from sub-optimal policies.

### 3.4 PERTURBED QUANTILE REGRESSION FOR DEEP Q-NETWORK

We propose a perturbed quantile regression (PQR) that is a practical algorithm for distributional reinforcement learning. We model a return distribution using quantile regression and update the quantile model by minimizing the objective function (1) induced by PDBOO. To compute the target distribution of (1), we propose a sampling method of $\xi$ from ambiguity set $\mathcal{U}_\Delta$. Since we employ a quantile model, sampling a reweight function $\xi$ can be reduced into sampling an $N$-dimensional weight vector $\boldsymbol{\xi} := [\xi_1, \cdots, \xi_N]$ where $\sum_{i=1}^{N}\xi_i = N$ and $\xi_i \geq 0$ for all $i \in \{1, \cdots, N\}$. Based on the QR-DQN setup, note that the condition $\int_{w \in \Omega}\xi(w)\mathbb{P}(w)dw = 1$ turns into $\sum_{i=1}^{N}\frac{1}{N}\xi_i = 1$ since the quantile level is set as $\tau_i = \frac{i}{N}$.

A key issue is how to construct an ambiguity set with bound $\Delta_t$ and then sample $\boldsymbol{\xi}$. A natural class of distribution for practical use is the *symmetric Dirichlet distribution* with concentration $\boldsymbol{\beta}$, which

---

**Algorithm 1:** Perturbed QR-DQN (PQR)

---

**Input:** transition $(s, a, r, s')$, discount $\gamma \in [0, 1)$, timestep $t > 0$, epsilon $\epsilon > 0$, concentration $\boldsymbol{\beta}$

    Initialize $\Delta_0 > 0$

    $\Delta_t \leftarrow \Delta_0 t^{-(1+\epsilon)}$

    $\boldsymbol{\xi} \leftarrow \max\left(\mathbf{1}^N + \Delta_t(N\boldsymbol{x} - \mathbf{1}^N), 0\right)$ where $\boldsymbol{x} \sim \text{Dir}(\boldsymbol{\beta})$    # Sample $\xi \sim \bar{\mathcal{U}}_{\Delta_t}(Z^{(t)})$

    $\boldsymbol{\xi} \leftarrow N\boldsymbol{\xi}/\sum\xi_i$

    $a^* \leftarrow \arg\max_{a'}\mathbb{E}_\xi[Z(s', a')]$    # Select greedy action with distorted expectation

    $\mathcal{T}\theta_j \leftarrow r + \gamma\theta_j(s', a^*), \quad \forall j$

    $t \leftarrow t + 1$

**Output:** $\sum_{i=1}^{N}\mathbb{E}_j[\rho_{\hat{\tau}_i}^\kappa(\mathcal{T}\theta_j - \theta_i(s, a))]$

---

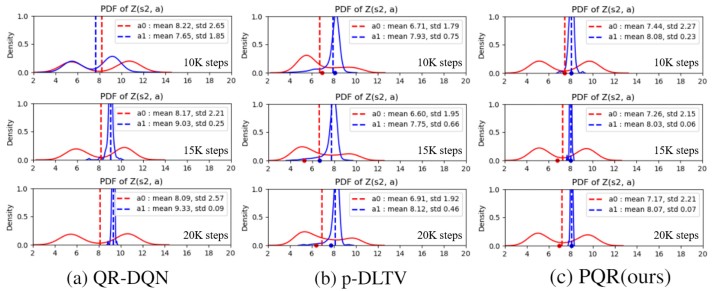

Figure 3: **(Left)** Empirical return distribution plot in N-Chain environment. Since QR-DQN does not depend on other criterion, the dots are omitted. **(Right)** Mean and standard-deviation difference between each algorithm and ground truth distribution.

represents distribution over distributions. (i.e. $x \sim \text{Dir}(\boldsymbol{\beta})$.) If $\beta$ is small, then, most of the mass is concentrated on a few elements. Otherwise, all elements are similar to each other and produce evenly distributed weight. By using the Dirichlet distribution, we sample a random vector, $x \sim \text{Dir}(\boldsymbol{\beta})$, and define the reweight distribution as $\boldsymbol{\xi} := \mathbf{1}^N + \alpha(N x - \mathbf{1}^N)$. From the construction of $\boldsymbol{\xi}$, we have $1 - \alpha \le \xi_i \le 1 + \alpha(N-1)$ for all $i$ and it follows that $|1 - \xi_i| \le \alpha(N-1)$ for all $i$. By controlling $\alpha$, we can bound the deviation of $\xi_i$ from 1 and bound the perturbation gap as

$$\sup_{s,a} |\mathbb{E}[Z(s,a)] - \mathbb{E}_\xi[Z(s,a)]| = \sup_{s,a} \left| \int_{w \in \Omega} Z(w; s,a)(1 - \xi(w))\mathbb{P}(w)dw \right|$$

$$\le \sup_{w \in \Omega} |1 - \xi(w)| \sup_{s,a} \mathbb{E}[|Z(s,a)|] \le \sup_{w \in \Omega} |1 - \xi(w)|V_{\max} \le \alpha(N-1)V_{\max}.$$

Hence, letting $\alpha \le \frac{\Delta}{(N-1)V_{\max}}$ is sufficient to obtain $d(Z; \xi) \le \Delta$ in the quantile setting. We set $\beta = 0.05 \cdot \mathbf{1}^N$ to generate a constructive perturbation $\xi_n$ which gap is close to the bound $\Delta_n$. To satisfy the condition stated in Theorem 4, we set $\Delta_t = \Delta_0 t^{-(1+\epsilon)}$ where $\Delta_0$ is a hyperparameter. The detailed procedure of the proposed method is summarized in Algorithm 1.

## 4 EXPERIMENTAL RESULTS AND DETAILS

In this section, we evaluate the performance on N-Chain, LunarLander and Atari games to demonstrate the efficiency of randomized criteria, comparing with QR-DQN ($\epsilon$-greedy) and DLTV (OFU).

**N-Chain.**    As the mean of each return is designed to be similar, it is useful to examine the learning behavior of the empirical return distribution for each algorithm. Figure 3 shows the empirical PDF of return distribution by using Gaussian kernel density estimation. PQR estimates the ground truth much better than other baselines with much closer mean and standard-deviation. Although the optimal policy was performed, QR-DQN overestimates the optimal Q-value of $(s_2, a_1)$ as $\hat{\mu} = 9.33$ while the ground truth is computed as $\mu = 10\gamma^2 = 8.1$. PQR estimated the target relatively better compared to QR-DQN as the agent often chooses an alternative action from a risk-neutral perspective. Only by chainging from optimism to a randomized scheme, p-DLTV made a much better estimate than DLTV.

**LunarLander-v2.**    Next, we evaluate the performance on LunarLander-v2 environment where its input is given by 8-dimensional coordinates with 4 different discrete actions. The goal of the agent is to reach the landing pad with a given threshold of 200 points without crashing. Without obtaining the extra point of +100, the agent cannot reach the threshold.

Figure 4(a) shows that p-DLTV and PQR have reached the threshold 200 points faster than the other two algorithms. Significantly, p-DLTV successfully reached the goal just by randomizing the criterion from OFU while DLTV did not learn to obtain the extra point. The result shows that one-sided tendency on risk leads to learning failure of an agent. We note that the randomized risk criterion could alleviate the stuckness within an affordable time budget.

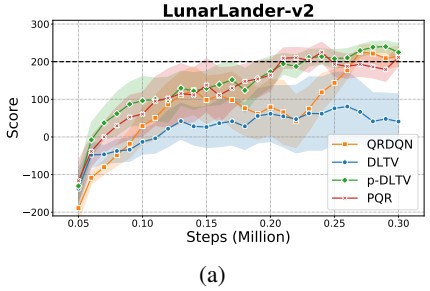

| | Hitting Time (M) |
|---|---|
| QR-DQN | 0.23 |
| DLTV | N.A |
| p-DLTV | 0.18 |
| PQR(ours) | **0.16** |

(a)                                                                              (b)

Figure 4: (a) Evaluation curves on LunarLander-v2. All curves are the average of three random seeds and the shaded area represents the standard deviation. We smoothed the curve over 5 consecutive steps. (b) Hitting time to reach a given threshold.

**Atari Games.**   Lastly, we evaluate the performance in Atari games, including multiple environments, each of which contained intrinsic uncertainty in different ways.

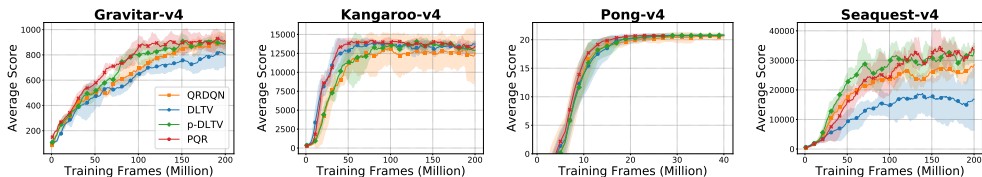

Figure 5: Evaluation curves on Atari games. We smoothed all curves over 10 consecutive steps with three random seeds. In case of Pong-v4, we resize the x-axis, since it can easily obtain the optimal policy with few interactions due to its environmental simplicity.

We plot the average performance of each algorithm in the evaluation step on 200M frames with 3 random seeds. In Figure 5, PQR achieves the shortest time to hit the asymptotic highest performance in various Atari games. Moreover, our proposed method is stable with relatively low variance in all four environments. As the bound of perturbation gap converges to 0, the plausible action is chosen from the shrinking ambiguity set, and thus the agent gradually performs only the optimal policy. Further experimental details are in Appendix D

In all experiments, just adding randomness to the coefficient $c_t$ shows the significant improvement supporting that the randomized risk criterion was superior to OFU in distributional RL. In the majority of environments where intrinsic uncertainties exist, OFU has difficulty in making a decision that matches risk-neutral purpose, because two uncertainties are intertwined during learning. We observe that PQR outperforms the baseline by showing faster convergence without degrading the performance. All experimental results support that the randomizing scheme is more effective than OFU in distributional RL.

## 5   CONCLUSIONS

In this paper, we proposed a general framework of risk-based exploration which captures the characteristics of a return distribution. Without resorting to a pre-defined risk criterion, we revealed and resolved the stuckness where one-sided tendency on risk can lead to biased action selection. To our best knowledge, this paper is the first attempt in DRL to integrate risk-sensitivity and exploration by using time-varying Bellman objective with theoretical analysis. We prove that PDBOO is theoretically guaranteed for convergence, and has a unique fixed point in a weaker condition than contraction. By obtaining the same fixed point as the standard Bellman optimality operator, we can obtain a risk-neutral optimal policy even if the agent explored with distortion risk measure. Our method shows better empirical results compared to recent exploration methods for distributional RL with various environments.

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

# A  PROOF

## A.1  TECHNICAL LEMMA

Before proving our theoretical results, we present two inequalities for supremum to clear the description.

1. $\sup\limits_{x\in X}|f(x) + g(x)| \leq \sup\limits_{x\in X}|f(x)| + \sup\limits_{x\in X}|g(x)|$

2. $\left|\sup\limits_{x\in X} f(x) - \sup\limits_{x'\in X} g(x')\right| \leq \sup\limits_{x,x'\in X} |f(x) - g(x')|$

*Proof of 1.* Since $|f(x) + g(x)| \leq |f(x)| + |g(x)|$ holds for all $x \in X$,

$$\sup\limits_{x\in X} |f(x) + g(x)| \leq \sup\limits_{x\in X}(|f(x)| + |g(x)|) \leq \sup\limits_{x\in X} |f(x)| + \sup\limits_{x\in X} |g(x)|$$

∎

*Proof of 2.* Since $\left|\|a\| - \|b\|\right| \leq \|a - b\|$ for any norm $\|\cdot\|$ and for a large enough $M$,

$$\sup\limits_{x,x'\in X} |f(x) - g(x')| \geq \sup\limits_{x\in X}|f(x) - g(x)| = \sup\limits_{x\in X}|(f(x) + M) - (g(x) + M)|$$

$$\geq \left|\sup\limits_{x\in X}(f(x) + M) - \sup\limits_{x\in X}(g(x) + M)\right|$$

$$= \left|\sup\limits_{x\in X} f(x) - \sup\limits_{x'\in X} g(x')\right|$$

∎

## A.2  PROOF OF THEOREM 3

**Theorem 3.** If $\xi_t$ converges to 1 uniformly on $\Omega$, then $\mathbb{E}\mathcal{T}_{\xi_t}$ also converges to $\mathbb{E}\mathcal{T}$ uniformly on $\mathcal{Z}$ for all $s \in \mathcal{S}$ and $a \in \mathcal{A}$.

*Proof.* Recall that $\mathcal{Z} = \left\{Z : \mathcal{S} \times \mathcal{A} \to \mathscr{P}(\mathbb{R})\,\big|\, \mathbb{E}[|Z(s,a)|] \leq V_{\max}, \forall (s,a)\right\}$. Then for any $Z \in \mathcal{Z}$ and $\xi \in \Xi$,

$$\mathbb{E}[|\mathcal{T}_\xi Z|] \leq R_{\max} + \gamma \frac{R_{\max}}{1-\gamma} = \frac{R_{\max}}{1-\gamma} = V_{\max}.$$

which implies PDBOO is closed in $\mathcal{Z}$, i.e. $\mathcal{T}_\xi Z \in \mathcal{Z}$ for all $\xi \in \Xi$. Hence, for any sequence $\xi_t$, $Z^{(n)} = \mathcal{T}_{\xi_{n:1}} Z \in \mathcal{Z}$ for any $n \geq 0$.

Since $\xi_t$ converges to 1 uniformly on $\Omega$, there exists $T$ such that for any $t > T$,

$$\sup\limits_{w\in\Omega} |\xi_t(w) - 1| \leq \epsilon.$$

For any $Z \in \mathcal{Z}$, $s \in \mathcal{S}, a \in \mathcal{A}$, and $t > T$, by using Hölder's inequality,

$$\sup\limits_{Z\in\mathcal{Z}} \sup\limits_{s,a} |\mathbb{E}_{\xi_t}[Z(s,a)] - \mathbb{E}[Z(s,a)]| = \sup\limits_{Z\in\mathcal{Z}} \sup\limits_{s,a} \left|\int_{w\in\Omega} (1 - \xi_t(w))Z(s,a,w)\mathbb{P}(w)dw\right|$$

$$\leq \sup\limits_{w\in\Omega} |\xi_t(w) - 1| \sup\limits_{Z\in\mathcal{Z}} \sup\limits_{s,a} \left|\int_{w\in\Omega} |Z(s,a,w)|\mathbb{P}(w)dw\right|$$

$$\leq \epsilon V_{\max}$$

which implies that $\mathbb{E}_{\xi_t}$ converges to $\mathbb{E}$ uniformly on $\mathcal{Z}$ for all $s, a$.

By using A.1, we can get the desired result.

$$\sup_{Z\in\mathcal{Z}}\sup_{s,a}|\mathbb{E}[\mathcal{T}_{\xi_t}Z(s,a)]-\mathbb{E}[\mathcal{T}Z(s,a)]|$$

$$\leq \sup_{Z\in\mathcal{Z}}\sup_{s,a}|\mathbb{E}[\mathcal{T}_{\xi_t}Z(s,a)]-\mathbb{E}_{\xi_t}[\mathcal{T}_{\xi_t}Z(s,a)]|+\sup_{Z\in\mathcal{Z}}\sup_{s,a}|\mathbb{E}_{\xi_t}[\mathcal{T}_{\xi_t}Z(s,a)]-\mathbb{E}[\mathcal{T}Z(s,a)]|$$

$$\leq \epsilon V_{\max}+\gamma\sup_{Z\in\mathcal{Z}}\sup_{s,a}\mathbb{E}_{s'}\left[\left|\sup_{a'}\mathbb{E}_{\xi_t}[Z(s',a')]-\sup_{a''}\mathbb{E}[Z(s',a'')]\right|\right]$$

$$\leq \epsilon V_{\max}+\gamma\sup_{Z\in\mathcal{Z}}\sup_{s',a'}|\mathbb{E}_{\xi_t}[Z(s',a')]-\mathbb{E}[Z(s',a')]|$$

$$\leq \epsilon V_{\max}+\gamma\epsilon V_{\max}$$

$$= (1+\gamma)\epsilon V_{\max}.$$

∎

### A.3 PROOF OF THEOREM 4

**Theorem 4.** For a sequence of bound $\Delta_n$, let $\bar{\mathcal{U}}_{\Delta_n}(Z^{(n-1)}):=\bigcap_{s,a}\mathcal{U}_{\Delta_n}(Z^{(n-1)}(s,a))$. If we sample $\xi_n$ from $\bar{\mathcal{U}}_{\Delta_n}(Z^{(n-1)})$ for every iteration and $\sum_{n=1}^{\infty}\Delta_n<\infty$ holds, then, the expectation of iterated operator $\mathcal{T}_{\xi_{n:1}}$ has a fixed point $\mathbb{E}[Z^*]$. Moreover, the following bound holds,

$$\sup_{s,a}\left|\mathbb{E}[Z^{(n)}(s,a)]-\mathbb{E}[Z^*(s,a)]\right|\leq\sum_{k=n}^{\infty}\left(2\gamma^{k-1}V_{\max}+2\sum_{i=1}^{k}\gamma^i(\Delta_{k+2-i}+\Delta_{k+1-i})\right).$$

*Proof.* We denote $a_i^*(\xi_n)=\arg\max_{a'}\mathbb{E}_{\xi_n}[Z_i^{(n-1)}(s',a')]$ as the greedy action of $Z_i^{(n-1)}$ under perturbation $\xi_n$. Also, we denote $\sup_{s,a}|\cdot|$ which is the supremum norm over $s$ and $a$ as $\|\cdot\|_{sa}$.

Before we start from the term $\left\|\mathbb{E}[Z^{(k+1)}]-\mathbb{E}[Z^{(k)}]\right\|_{sa}$, for a given $(s,a)$,

$$\left|\mathbb{E}[Z^{(k+1)}(s,a)]-\mathbb{E}[Z^{(k)}(s,a)]\right|$$

$$\leq \gamma\sup_{s'}\left|\mathbb{E}[Z^{(k)}(s',a^*(\xi_{k+1}))]-\mathbb{E}[Z^{(k-1)}(s',a^*(\xi_k))]\right|$$

$$\leq \gamma\sup_{s'}\left(\left|\mathbb{E}[Z^{(k)}(s',a^*(\xi_{k+1}))]-\max_{a'}\mathbb{E}[Z^{(k)}(s',a')]\right|+\left|\max_{a'}\mathbb{E}[Z^{(k)}(s',a')]\right.\right.$$

$$\left.\left.-\max_{a'}\mathbb{E}[Z^{(k-1)}(s',a')]\right|+\left|\max_{a'}\mathbb{E}[Z^{(k-1)}(s',a')]-\mathbb{E}[Z^{(k-1)}(s',a^*(\xi_k))]\right|\right)$$

$$\leq \gamma\sup_{s',a'}\left|\mathbb{E}[Z^{(k)}(s',a')]-\mathbb{E}[Z^{(k-1)}(s',a')]\right|+\gamma\sum_{i=k-1}^{k}\sup_{s'}\left(\left|\mathbb{E}[Z^{(i)}(s',a^*(\xi_{i+1}))]\right.\right.$$

$$\left.\left.-\max_{a'}\mathbb{E}[Z^{(i)}(s',a')]\right|\right)$$

$$\leq \gamma\left\|\mathbb{E}[Z^{(k)}]-\mathbb{E}[Z^{(k-1)}]\right\|_{sa}+\gamma\sum_{i=k-1}^{k}\sup_{s'}\left(\left|\mathbb{E}[Z^{(i)}(s',a^*(\xi_{i+1}))]\right.\right.$$

$$\left.\left.-\mathbb{E}_{\xi_{i+1}}[Z^{(i)}(s',a^*(\xi_{i+1}))]\right|+\left|\max_{a'}\mathbb{E}_{\xi_{i+1}}[Z^{(i)}(s',a')]-\max_{a''}\mathbb{E}[Z^{(i)}(s',a'')]\right|\right)$$

$$\leq \gamma\left\|\mathbb{E}[Z^{(k)}]-\mathbb{E}[Z^{(k-1)}]\right\|_{sa}+2\gamma\sum_{i=k-1}^{k}\sup_{s',a'}\left(\left|\mathbb{E}[Z^{(i)}(s',a')]-\mathbb{E}_{\xi_{i+1}}[Z^{(i)}(s',a')]\right|\right)$$

$$\leq \gamma\left\|\mathbb{E}[Z^{(k)}]-\mathbb{E}[Z^{(k-1)}]\right\|_{sa}+2\gamma\sum_{i=k-1}^{k}\Delta_{i+1}$$

where we use A.1.1 in third and fifth line and A.1.2 in sixth line.

Taking a supremum over $s$ and $a$, then for all $k > 0$,

$$
\left\| \mathbb{E}[Z^{(k+1)}] - \mathbb{E}[Z^{(k)}] \right\|_{sa} \leq \gamma \left\| \mathbb{E}[Z^{(k)}] - \mathbb{E}[Z^{(k-1)}] \right\|_{sa} + 2 \sum_{i=k-1}^{k} \gamma \Delta_{i+1}
$$

$$
\leq \gamma^2 \left\| \mathbb{E}[Z^{(k-1)}] - \mathbb{E}[Z^{(k-2)}] \right\|_{sa} + 2 \sum_{i=k-2}^{k-1} \gamma^2 \Delta_{i+1} + 2 \sum_{i=k-1}^{k} \gamma \Delta_{i+1}
$$

$$
\vdots
$$

$$
\leq \gamma^k \left\| \mathbb{E}[Z^{(1)}] - \mathbb{E}[Z] \right\|_{sa} + 2 \sum_{i=1}^{k} \gamma^i (\Delta_{k+2-i} + \Delta_{k+1-i})
$$

$$
\leq 2\gamma^k V_{\max} + 2 \sum_{i=1}^{k} \gamma^i (\Delta_{k+2-i} + \Delta_{k+1-i})
$$

Since $\sum_{i=1}^{\infty} \gamma^i = \frac{\gamma}{1-\gamma} < \infty$ and $\sum_{i=1}^{\infty} \Delta_i < \infty$ by assumption, we have

$$
\sum_{i=1}^{k} \gamma^i \Delta_{k+1-i} \to 0
$$

which is resulted from the convergence of Cauchy product of two sequences $\{\gamma^i\}$ and $\{\Delta_i\}$. Hence, $\{\mathbb{E}[Z^{(k)}]\}$ is a Cauchy sequence and therefore converges for every $Z \in \mathcal{Z}$.

Let $\mathbb{E}[Z^*]$ be the limit point of the sequence $\{\mathbb{E}[Z^{(n)}]\}$. Then,

$$
\left\| \mathbb{E}[Z^*] - \mathbb{E}[Z^{(n)}] \right\|_{sa} = \lim_{l \to \infty} \left\| \mathbb{E}[Z^{(n+l)}] - \mathbb{E}[Z^{(n)}] \right\|_{sa}
$$

$$
\leq \sum_{k=n}^{\infty} \left\| \mathbb{E}[Z^{(k+1)}] - \mathbb{E}[Z^{(k)}] \right\|_{sa}
$$

$$
= \sum_{k=n}^{\infty} \left( 2\gamma^k V_{\max} + 2 \sum_{i=1}^{k} \gamma^i (\Delta_{k+2-i} + \Delta_{k+1-i}) \right).
$$

∎

### A.4 PROOF OF THEOREM 5

**Theorem 5.** If $\{\Delta_n\}$ follows the assumption in Theorem 4, then $\mathbb{E}[Z^*]$ is the unique solution of Bellman optimality equation.

*Proof.* The proof follows by linearity of expectation. Denote the Q-value based operator as $\bar{\mathcal{T}}$. Note that $\Delta_n$ converges to 0 with regularity of $\mathcal{Z}$ implies that $\xi_n$ converges to 1 uniformly on $\Omega$. By Theorem 3, for a given $\epsilon > 0$, there exists a constant $K = \max(K_1, K_2)$ such that for every $k \geq K_1$,

$$
\sup_{Z \in \mathcal{Z}} \| \bar{\mathcal{T}}_{\xi_k} \mathbb{E}[Z] - \bar{\mathcal{T}} \mathbb{E}[Z] \|_{sa} \leq \frac{\epsilon}{2}.
$$

Since $\bar{\mathcal{T}}$ is continuous, for every $k \geq K_2$,

$$
\| \bar{\mathcal{T}} \mathbb{E}[Z^{(k)}] - \bar{\mathcal{T}} \mathbb{E}[Z^*] \|_{sa} \leq \frac{\epsilon}{2}.
$$

Thus, it holds that

$$
\| \bar{\mathcal{T}}_{\xi_{k+1}} \mathbb{E}[Z^{(k)}] - \bar{\mathcal{T}} \mathbb{E}[Z^*] \|_{sa} \leq \| \bar{\mathcal{T}}_{\xi_{k+1}} \mathbb{E}[Z^{(k)}] - \bar{\mathcal{T}} \mathbb{E}[Z^{(k)}] \|_{sa} + \| \bar{\mathcal{T}} \mathbb{E}[Z^{(k)}] - \bar{\mathcal{T}} \mathbb{E}[Z^*] \|_{sa}
$$

$$
\leq \sup_{Z \in \mathcal{Z}} \| \bar{\mathcal{T}}_{\xi_{k+1}} \mathbb{E}[Z] - \bar{\mathcal{T}} \mathbb{E}[Z] \|_{sa} + \| \bar{\mathcal{T}} \mathbb{E}[Z^{(k)}] - \bar{\mathcal{T}} \mathbb{E}[Z^*] \|_{sa}
$$

$$
\leq \frac{\epsilon}{2} + \frac{\epsilon}{2}
$$

$$
= \epsilon.
$$

Therefore, we have
$$\mathbb{E}[Z^*] = \lim_{k \to \infty} \mathbb{E}[Z^{(k)}] = \lim_{k \to \infty} \mathbb{E}[Z^{(k+1)}] = \lim_{k \to \infty} \mathbb{E}[\mathcal{T}_{\xi_{k+1}} Z^{(k)}] = \lim_{k \to \infty} \bar{\mathcal{T}}_{\xi_{k+1}} \mathbb{E}[Z^{(k)}] = \bar{\mathcal{T}} \mathbb{E}[Z^*]$$
Since the standard Bellman optimality operator has a unique solution, we derived the desired result. ∎

## B    ALGORITHM PIPELINE

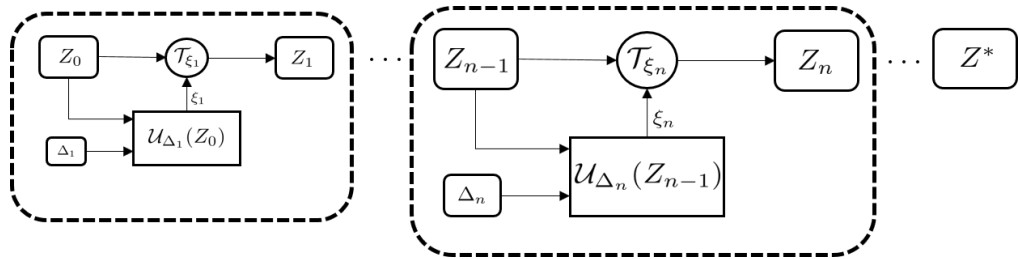

Figure 6: Pipeline of PDBOO.

Figure 6 shows the pipeline of our algorithm. With the schedule of perturbation bound $\{\Delta_n\}$, the ambiguity set $\mathcal{U}_{\Delta_n}(Z_{n-1})$ can be defined by previous $Z_{n-1}$. For each step, (distributional) perturbation $\xi_n$ is sampled from $\mathcal{U}_{\Delta_n}(Z_{n-1})$ by the symmetric Dirichlet distribution and then PDBOO $\mathcal{T}_{\xi_n}$ can be performed.

## C    IMPLEMENTATION DETAILS

Except for each own hyperparameter, our algorithms and DLTV shares the same hyperparameter and network architecture with QR-DQN. Also, we set up p-DLTV by only multiplying a gaussian noise $\mathcal{N}(0,1)$ to the coefficient of DLTV. We do not combine any additional improvements such as double Q-learning, dueling network, prioritized replay, and $n$-step update. Experiments on LunarLander-v2 and Atari games were performed with 3 random seeds.

### C.1    N-CHAIN

For hyperparameter settings, we initialize all agents with a random policy for 500 steps and then train by 20K steps with 10 random seeds. $\epsilon-$greedy policy which is only executed on QR-DQN annealed linearly from 1 to 0.01 over the first 2500 steps. The batch size was 64 and the discount factor was $\gamma = 0.9$. We update the network every 1 step and the number of steps to update targets was 25 steps.

### C.2    LUNARLANDER-V2

The hyperparameters of QR-DQN were followed by the settings reported in Raffin et al. (2019) for a fair comparison. Our experiments used 2 layers of MLP with 256 hidden units. We used the experience replay with batch size 128 and the buffer size $1 \times 10^5$. The number of quantiles $N$ was 170 and $\gamma = 0.995$. As a stochastic gradient optimizer, we adopt Adam with a learning rate $1.5 \times 10^{-3}$ with a linear decaying schedule. $\epsilon$-greedy schedule was only performed on QR-DQN. For the rest, we used $c_t = 0.05$, and $\Delta = 5 \times 10^4$. We evaluated each algorithm for every 10K training steps by averaging 5 episodes.

### C.3    ATARI GAMES

For a fair comparison, our hyperparameter setting is aligned with Dabney et al. (2018b). The number of quantile fraction $N$ is 200. We set $\gamma = 0.99$ and $c_t = 50$ which refers to Mavrin et al. (2019) and $\Delta = 1 \times 10^6$. We use $\epsilon$-greedy with threshold $\epsilon = 0.01$ at training stage and $\epsilon = 0.001$ at test stage.

## C.4 PSEUODOCODE OF P-DLTV

---

**Algorithm 2:** Perturbed DLTV (p-DLTV)

---

**Input:** $(s, a, r, s'), \gamma \in [0, 1)$
$\quad Q(s', a') = \frac{1}{N} \sum_j \theta_j(s', a')$
$\quad c_t \sim c\, \mathcal{N}(0, \frac{\ln t}{t}) \quad$ # Randomize the coefficient
$\quad a^* \leftarrow \arg\max_{a'} (Q(s', a') + c_t \sqrt{\sigma_+^2(s', a')})$
$\quad \mathcal{T}\theta_j \leftarrow r + \gamma\theta_j(s', a^*), \quad \forall j$
**Output:** $\sum_{i=1}^N \mathbb{E}_j[\rho_{\hat\tau_i}^\kappa(\mathcal{T}\theta_j - \theta_i(s, a))]$

---

## D  FURTHER EXPERIMENTAL RESULTS OF ATARI GAMES

We test our algorithm under 30 no-op settings to align with previous works. We compare our baseline results with results from the DQN Zoo framework (Quan & Ostrovski, 2020), which provides the full benchmark results on 57 Atari games at 50M and 200M frames. We report the average of the best scores over 5 seeds for each baseline algorithms up to 50M frames. However, recent studies tried to follow the setting proposed by Machado et al. (2018) for reproducibility, where they recommended using sticky actions. Hence, we provide all human normalized scores results across 55 Atari games including previous report of Yang et al. (2019), Dopamine and DQN Zoo framework to help the follow-up researchers as a reference. We exclude *Defender* and *Surround* which is not reported on Yang et al. (2019) because of relialbility issues in the Dopamine framework.

|  | Mean | Median |
|---|---|---|
| **DQN-dopamine(50M)** | 401% | 51% |
| **DQN-zoo(50M)** | 507% | 69% |
| **DQN-zoo(200M)** | 804% | 84% |
| **DQN(200M)** | 221% | 79% |
| **QR-DQN-dopamine(50M)** | 562% | 93% |
| **QR-DQN-zoo(50M)** | 707% | 119% |
| **QR-DQN-zoo(200M)** | 1714% | 174% |
| **QR-DQN(200M)** | 902% | 193% |
| **IQN-dopamine(50M)** | 940% | 124% |
| **IQN-zoo(50M)** | 1057% | 143% |
| **IQN-zoo(200M)** | 2070% | 229% |
| **IQN(200M)** | 1112% | 218% |
| **RAINBOW-dopamine(50M)** | 965% | 123% |
| **RAINBOW-zoo(50M)** | 1366% | 154% |
| **RAINBOW-zoo(200M)** | 2115% | 246% |
| **RAINBOW(200M)** | 1213% | 230% |
| **PQR(50M)** | 1121% | 124% |

Table 1: Mean and median of best scores across 55 Atari games on 50M frames, measured as percentages of human baseline (Castro et al., 2018; Yang et al., 2019; Quan & Ostrovski, 2020)

Table 1 provides the mean and median human normalized scores across 55 Atari games. Due to the high computational cost, our algorithm was evaluated on 50M frames to provide results over as many environments as possible. It is observed that PQR shows better performance in terms of both mean and median metrics than QR-DQN. Since our method is based on QR-DQN, we would expect that PDBOO can be combined with IQN (Dabney et al., 2018a) or techniques in Rainbow (Hessel et al., 2018) as an efficient exploration method, and the performance can be further improved. Since ourmethod is based on QR-DQN, PDBOO can be combined with IQN (Dabney et al., 2018a) or the techniques in Rainbow (Hessel et al., 2018), such as double q-learning, n-step updates, dueling networks and prioritized experience replay. We would expect that PQR for efficient exploration would benefit from the additional improvement of performance as IQN and Rainbow outperform QR-DQN.

| GAMES | RANDOM | HUMAN | DQN(50M) | QR-DQN(50M) | IQN(50M) | RAINBOW(50M) | PQR(50M) |
|---|---|---|---|---|---|---|---|
| Alien | 227.8 | 7127.7 | 1633.9 | 1891.7 | 1991.0 | 4507.8 | 2455.8 |
| Amidar | 5.8 | 1719.5 | 344.6 | 761.2 | 853.6 | 2649.4 | 938.4 |
| Assault | 222.4 | 742.0 | 3744.7 | 13951.4 | 18910.2 | 11921.0 | 10759.2 |
| Asterix | 210.0 | 8503.3 | 5994.6 | 21225.7 | 33625.8 | 35954.1 | 10490.5 |
| Asteroids | 719.1 | 47388.7 | 1590.9 | 2113.3 | 2305.9 | 1917.8 | 1662.0 |
| Atlantis | 12850.0 | 29028.1 | 383243.6 | 962130.0 | 907430.0 | 954790.0 | 897640.0 |
| BankHeist | 14.2 | 753.1 | 477.1 | 1267.9 | 1298.1 | 1089.8 | 1038.8 |
| BattleZone | 2360.0 | 37187.5 | 22167.1 | 31242.0 | 33382.4 | 36797.6 | 28470.5 |
| BeamRider | 363.9 | 16926.5 | 8276.5 | 14710.0 | 23600.8 | 17280.9 | 10224.9 |
| Berzerk | 123.7 | 2630.4 | 644.5 | 936.0 | 960.9 | 2191.4 | 137873.1 |
| Bowling | 23.1 | 160.7 | 49.2 | 58.5 | 63.3 | 64.9 | 86.9 |
| Boxing | 0.1 | 12.1 | 90.2 | 99.7 | 99.6 | 99.9 | 97.1 |
| Breakout | 1.7 | 30.5 | 350.0 | 478.6 | 556.8 | 349.4 | 380.3 |
| Centipede | 2090.9 | 12017.0 | 6912.0 | 9334.4 | 8124.0 | 6055.8 | 7291.2 |
| ChopperCommand | 811.0 | 7387.8 | 1081.7 | 1453.6 | 2054.5 | 6277.6 | 1300.0 |
| CrazyClimber | 10780.5 | 35829.4 | 109112.5 | 111783.3 | 134466.5 | 168808.9 | 84390.9 |
| DemonAttack | 152.1 | 1971.0 | 9695.5 | 116973.3 | 115749.8 | 98203.8 | 73794.0 |
| DoubleDunk | -18.6 | -16.4 | -14.9 | -7.0 | -7.6 | 0.2 | -7.5 |
| Enduro | 0.0 | 860.5 | 850.5 | 2238.6 | 2345.7 | 2350.7 | 2341.2 |
| FishingDerby | -91.7 | -38.7 | 15.2 | 33.4 | 27.0 | 38.5 | 31.7 |
| Freeway | 0.0 | 29.6 | 24.6 | 33.5 | 33.8 | 34.0 | 34.0 |
| Frostbite | 65.2 | 4334.7 | 614.8 | 4346.2 | 6187.6 | 9902.2 | 4148.2 |
| Gopher | 257.6 | 2412.5 | 4289.8 | 7425.6 | 34520.2 | 37955.5 | 47054.5 |
| Gravitar | 173.0 | 3351.4 | 271.2 | 617.6 | 535.9 | 2253.4 | 635.8 |
| Hero | 1027.0 | 30826.4 | 10821.5 | 11823.4 | 13628.9 | 38189.0 | 12579.2 |
| IceHockey | -11.2 | 0.9 | -4.1 | -1.0 | -2.6 | 2.5 | -1.4 |
| Jamesbond | 29.0 | 302.8 | 524.3 | 1411.5 | 1336.8 | 15191.8 | 2121.8 |
| Kangaroo | 52.0 | 3035.0 | 8146.9 | 14885.9 | 14504.8 | 14670.3 | 14617.1 |
| Krull | 1598.0 | 2665.5 | 10626.8 | 11004.2 | 10028.9 | 9055.3 | 9746.1 |
| KungFuMaster | 258.5 | 22736.3 | 25251.6 | 37140.8 | 44248.7 | 32520.8 | 43258.6 |
| MontezumaRevenge | 0.0 | 4753.3 | 0.4 | 0.0 | 0.2 | 80.3 | 0.0 |
| MsPacman | 307.3 | 6951.6 | 2427.6 | 3385.0 | 3075.4 | 3894.8 | 2928.9 |
| NameThisGame | 2292.3 | 8049.0 | 7260.8 | 12595.6 | 12538.4 | 11609.4 | 10298.2 |
| Phoenix | 761.4 | 7242.6 | 12646.5 | 37830.1 | 37153.7 | 61410.6 | 20453.8 |
| Pitfall | -229.4 | 6463.7 | 0.0 | 0.0 | 0.0 | 0.0 | 0.0 |
| Pong | -20.7 | 14.6 | 19.2 | 21.0 | 21.0 | 21.0 | 21.0 |
| PrivateEye | 24.9 | 69571.3 | 542.7 | 100.0 | 100.0 | 160.0 | 372.4 |
| Qbert | 163.9 | 13455.0 | 6541.3 | 14885.8 | 15681.3 | 26067.9 | 15267.4 |
| Riverraid | 1338.5 | 17118.0 | 7467.2 | 10019.4 | 14311.3 | 17785.7 | 11175.3 |
| RoadRunner | 11.5 | 7845.0 | 32714.0 | 55506.2 | 55873.9 | 54265.5 | 50854.7 |
| Robotank | 2.2 | 11.9 | 35.7 | 56.6 | 55.5 | 67.6 | 60.3 |
| Seaquest | 68.4 | 42054.7 | 3239.3 | 8990.9 | 18943.3 | 3057.5 | 19652.5 |
| Skiing | -17098.1 | -4336.9 | -13445.9 | -9198.1 | -9310.5 | -14901.8 | -9299.3 |
| Solaris | 1236.3 | 12326.7 | 3388.0 | 2523.6 | 3612.9 | 3714.6 | 2640.0 |
| SpaceInvaders | 148.0 | 1668.7 | 1314.2 | 1960.3 | 3058.2 | 2746.3 | 1749.4 |
| StarGunner | 664.0 | 10250.0 | 30631.9 | 63950.3 | 81730.5 | 111760.9 | 62920.6 |
| Tennis | -23.8 | -8.3 | 0.0 | 0.0 | 0.0 | 0.0 | -1.0 |
| TimePilot | 3568.0 | 5229.2 | 3439.9 | 7835.8 | 6971.4 | 12731.9 | 6506.4 |
| Tutankham | 11.4 | 167.6 | 192.8 | 229.2 | 209.0 | 199.9 | 231.3 |
| UpNDown | 533.4 | 11693.2 | 11192.4 | 45653.5 | 65773.2 | 77524.5 | 36008.1 |
| Venture | 0.0 | 1187.5 | 154.2 | 17.5 | 54.1 | 1.1 | 993.3 |
| VideoPinball | 16256.9 | 17667.9 | 258626.0 | 182452.1 | 416941.4 | 589122.4 | 465578.3 |
| WizardOfWor | 563.5 | 4756.5 | 4914.8 | 19040.8 | 15892.2 | 13362.6 | 6132.8 |
| YarsRevenge | 3092.9 | 54576.9 | 18540.6 | 23385.8 | 23115.6 | 69092.9 | 27674.4 |
| Zaxxon | 32.5 | 9173.3 | 4890.3 | 9632.6 | 8686.6 | 23620.4 | 10806.6 |

Table 2: Raw scores for a single seed across all 55 games, starting with 30 no-op actions. We report the best scores for DQN, QR-DQN, IQN and Rainbow on 50M frames, averaged by 5 seeds. Reference values were provided by DQN Zoo framework (Quan & Ostrovski, 2020)

| GAMES | RANDOM | HUMAN | DQN(50M) | QR-DQN(50M) | IQN(50M) | RAINBOW(50M) | PQR(50M) |
|---|---|---|---|---|---|---|---|
| Alien | 227.8 | 7127.7 | 1688.1 | 2754.2 | 4016.3 | 2076.2 | 2455.8 |
| Amidar | 5.8 | 1719.5 | 888.2 | 841.6 | 1642.8 | 1669.6 | 938.4 |
| Assault | 222.4 | 742 | 1615.9 | 2233.1 | 4305.6 | 2535.9 | 10759.2 |
| Asterix | 210 | 8503.3 | 3326.1 | 3540.1 | 7038.4 | 5862.3 | 10490.5 |
| Asteroids | 719.1 | 47388.7 | 828.2 | 1333.4 | 1336.3 | 1345.1 | 1662.0 |
| Atlantis | 12850.0 | 29028.1 | 388466.7 | 879022.0 | 897558.0 | 870896.0 | 897640.0 |
| BankHeist | 14.2 | 753.1 | 720.2 | 964.1 | 1082.8 | 1104.9 | 1038.8 |
| BattleZone | 2360.0 | 37187.5 | 15110.3 | 25845.6 | 29959.7 | 32862.1 | 28470.5 |
| BeamRider | 343.9 | 16926.5 | 4771.3 | 7143.0 | 7113.7 | 6331.9 | 10224.9 |
| Berzerk | 123.7 | 2630.4 | 529.2 | 603.2 | 627.3 | 697.8 | 137873.1 |
| Bowling | 23.1 | 160.7 | 38.5 | 55.3 | 33.6 | 55.0 | 86.9 |
| Boxing | 0.1 | 12.1 | 80.0 | 96.6 | 97.8 | 96.3 | 97.1 |
| Breakout | 1.7 | 30.5 | 113.5 | 40.7 | 164.4 | 69.8 | 380.3 |
| Centipede | 2090.9 | 12017.0 | 3403.7 | 3562.5 | 3746.1 | 5087.6 | 7291.2 |
| ChopperCommand | 811 | 7387.8 | 1615.3 | 1600.3 | 6654.1 | 5982.0 | 1044.0 |
| CrazyClimber | 10780.5 | 35829.4 | 111493.8 | 108493.9 | 131645.8 | 135786.1 | 84390.9 |
| DemonAttack | 152.1 | 1971.0 | 4396.7 | 3182.6 | 7715.5 | 6346.4 | 73794.0 |
| DoubleDunk | -18.6 | -16.4 | -16.7 | 7.4 | 20.2 | 17.4 | -7.5 |
| Enduro | 0 | 860.5 | 799.5 | 2062.5 | 2268.1 | 2255.6 | 2341.2 |
| FishingDerby | -91.7 | -38.7 | 12.3 | 48.4 | 41.9 | 37.6 | 31.7 |
| Freeway | 0 | 29.6 | 25.8 | 33.5 | 33.5 | 33.2 | 34.0 |
| Frostbite | 65.2 | 4334.7 | 760.2 | 8022.8 | 7824.9 | 5697.2 | 4148.2 |
| Gopher | 257.6 | 2412.5 | 3495.8 | 3917.1 | 11192.6 | 7102.1 | 47054.5 |
| Gravitar | 173.0 | 3351.4 | 250.7 | 821.3 | 1083.5 | 926.2 | 635.8 |
| Hero | 1027 | 30826.4 | 12316.4 | 14980.1 | 18754.0 | 31845.8 | 12579.2 |
| IceHockey | -11.2 | 0.9 | -6.7 | -4.5 | 0.0 | 2.3 | -1.4 |
| Jamesbond | 29.0 | 302.8 | 500.0 | 802.3 | 1118.8 | 656.7 | 2121.8 |
| Kangaroo | 52.0 | 3035.0 | 6768.2 | 4727.3 | 11385.4 | 13133.1 | 14617.1 |
| Krull | 1598 | 2665.5 | 6181.1 | 8073.9 | 8661.7 | 6292.5 | 9746.1 |
| KungFuMaster | 258.5 | 22736.3 | 20418.8 | 20988.3 | 33099.9 | 26707.0 | 43258.6 |
| MontezumaRevenge | 0.0 | 4753.3 | 2.6 | 300.5 | 0.7 | 501.2 | 0.0 |
| MsPacman | 307.3 | 6951.6 | 2727.2 | 3313.9 | 4714.4 | 3406.4 | 2928.9 |
| NameThisGame | 2292.3 | 8049.0 | 5697.3 | 7307.9 | 9432.8 | 9389.5 | 10298.2 |
| Phoenix | 761.4 | 7245.6 | 5833.7 | 4641.1 | 5147.2 | 8272.9 | 20453.8 |
| Pitfall | -229.4 | 6463.7 | -16.8 | -3.4 | -0.4 | 0 | 0.0 |
| Pong | -20.7 | 14.6 | 13.2 | 19.2 | 19.9 | 19.4 | 21.0 |
| PrivateEye | 24.9 | 69571.3 | 1884.6 | 680.7 | 1287.3 | 4298.8 | 372.4 |
| Qbert | 163.9 | 13455.0 | 8216.2 | 17228.0 | 15045.5 | 17121.4 | 15267.4 |
| Riverraid | 1338.5 | 17118.0 | 9077.8 | 13389.4 | 14868.6 | 15748.9 | 11175.3 |
| RoadRunner | 11.5 | 7845.0 | 39703.1 | 44619.2 | 50534.1 | 51442.4 | 50854.7 |
| Robotank | 2.2 | 11.9 | 25.8 | 53.6 | 65.9 | 63.6 | 60.3 |
| Seaquest | 68.4 | 42054.7 | 1585.9 | 4667.9 | 20081.3 | 3916.2 | 19652.5 |
| Skiing | -17098.1 | -4336.9 | -17038.2 | -14401.6 | -13755.6 | -17960.1 | -9299.3 |
| Solaris | 1236.3 | 12326.7 | 2029.5 | 2361.7 | 2234.5 | 2922.2 | 2640.0 |
| SpaceInvaders | 148.0 | 1668.7 | 1361.1 | 940.2 | 3115.0 | 1908.0 | 1749.4 |
| StarGunner | 664.0 | 10250.0 | 1676.5 | 23593.3 | 60090.0 | 39456.3 | 62920.6 |
| Tennis | -23.8 | -9.3 | -0.1 | 19.2 | 3.5 | 0.0 | -1.0 |
| TimePilot | 3568.0 | 5229.2 | 3200.9 | 6622.8 | 9820.6 | 9324.4 | 6506.4 |
| Tutankham | 11.4 | 167.6 | 138.8 | 209.9 | 250.4 | 252.2 | 231.3 |
| UpNDown | 533.4 | 11693.2 | 10405.6 | 29890.1 | 44327.6 | 18790.7 | 36008.1 |
| Venture | 0 | 1187.5 | 50.8 | 1099.6 | 1134.5 | 1488.9 | 993.3 |
| VideoPinball | 16256.9 | 17667.9 | 216042.7 | 250650.0 | 486111.5 | 536364.4 | 465578.3 |
| WizardOfWor | 563.5 | 4756.5 | 2664.9 | 2841.8 | 6791.4 | 7562.7 | 6132.8 |
| YarsRevenge | 3092.9 | 54576.9 | 20375.7 | 66055.9 | 57960.3 | 31864.4 | 27674.4 |
| Zaxxon | 32.5 | 9173.3 | 1928.6 | 8177.2 | 12048.6 | 14117.5 | 10806.6 |

Table 3: Raw scores for a single seed across all 55 games. We report the best scores for DQN, QR-DQN, IQN, and Rainbow on 50M frames, averaged by 5 seeds. Reference values were provided by Dopamine framework (Castro et al., 2018).

| GAMES | RANDOM | HUMAN | DQN(200M) | QR-DQN(200M) | IQN(200M) | PQR(50M) |
|---|---|---|---|---|---|---|
| Alien | 227.8 | 7127.7 | 1620.0 | 4871.0 | 7022.0 | 2455.8 |
| Amidar | 5.8 | 1719.5 | 978.0 | 1641.0 | 2946.0 | 938.4 |
| Assault | 222.4 | 742.0 | 4280.4 | 22012.0 | 29091.0 | 10759.2 |
| Asterix | 210.0 | 8503.3 | 4359.0 | 261025.0 | 342016.0 | 10490.5 |
| Asteroids | 719.1 | 47388.7 | 1364.5 | 4226.0 | 2898.0 | 1662.0 |
| Atlantis | 12850.0 | 29028.1 | 279987.0 | 971850.0 | 978200.0 | 897640.0 |
| BankHeist | 14.2 | 753.1 | 455.0 | 1249.0 | 1416.0 | 1038.8 |
| BattleZone | 2360.0 | 37187.5 | 29900.0 | 39268.0 | 42244.0 | 28470.5 |
| BeamRider | 363.9 | 16926.5 | 8627.5 | 34821.0 | 42776.0 | 10224.9 |
| Berzerk | 123.7 | 2630.4 | 585.6 | 3117.0 | 1053.0 | 137873.1 |
| Bowling | 23.1 | 160.7 | 50.4 | 77.2 | 86.5 | 86.9 |
| Boxing | 0.1 | 12.1 | 88.0 | 99.9 | 99.8 | 97.1 |
| Breakout | 1.7 | 30.5 | 385.5 | 742.0 | 734.0 | 380.3 |
| Centipede | 2090.9 | 12017.0 | 4657.7 | 12447.0 | 11561.0 | 7291.2 |
| ChopperCommand | 811.0 | 7387.8 | 6126.0 | 14667.0 | 16836.0 | 1044.0 |
| CrazyClimber | 10780.5 | 35829.4 | 110763.0 | 161196.0 | 179082.0 | 84390.9 |
| DemonAttack | 152.1 | 1971.0 | 12149.4 | 121551.0 | 128580.0 | 73794.0 |
| DoubleDunk | -18.6 | -16.4 | -6.6 | 21.9 | 5.6 | -7.5 |
| Enduro | 0.0 | 860.5 | 729.0 | 2355.0 | 2359.0 | 2341.2 |
| FishingDerby | -91.7 | -38.7 | -4.9 | 39.0 | 33.8 | 31.7 |
| Freeway | 0.0 | 29.6 | 30.8 | 34.0 | 34.0 | 34.0 |
| Frostbite | 65.2 | 4334.7 | 797.4 | 4384.0 | 4324.0 | 4148.2 |
| Gopher | 257.6 | 2412.5 | 8777.4 | 113585.0 | 118365.0 | 47054.5 |
| Gravitar | 173.0 | 3351.4 | 473.0 | 995.0 | 911.0 | 635.8 |
| Hero | 1027.0 | 30826.4 | 20437.8 | 21395.0 | 28386.0 | 12579.2 |
| IceHockey | -11.2 | 0.9 | -1.9 | -1.7 | 0.2 | -1.4 |
| Jamesbond | 29.0 | 302.8 | 768.5 | 4703.0 | 35108.0 | 2121.8 |
| Kangaroo | 52.0 | 3035.0 | 7259.0 | 15356.0 | 15487.0 | 14617.1 |
| Krull | 1598.0 | 2665.5 | 8422.3 | 11447.0 | 10707.0 | 9746.1 |
| KungFuMaster | 258.5 | 22736.3 | 26059.0 | 76642.0 | 73512.0 | 43258.6 |
| MontezumaRevenge | 0.0 | 4753.3 | 0.0 | 0.0 | 0.0 | 0.0 |
| MsPacman | 307.3 | 6951.6 | 3085.6 | 5821.0 | 6349.0 | 2928.9 |
| NameThisGame | 2292.3 | 8049.0 | 8207.8 | 21890.0 | 22682.0 | 10298.2 |
| Phoenix | 761.4 | 7242.6 | 8485.2 | 16585.0 | 56599.0 | 20453.8 |
| Pitfall | -229.4 | 6463.7 | -286.1 | 0.0 | 0.0 | 0.0 |
| Pong | -20.7 | 14.6 | 19.5 | 21.0 | 21.0 | 21.0 |
| PrivateEye | 24.9 | 69571.3 | 146.7 | 350.0 | 200.0 | 372.4 |
| Qbert | 163.9 | 13455.0 | 13117.3 | 572510.0 | 25750.0 | 15267.4 |
| Riverraid | 1338.5 | 17118.0 | 7377.6 | 17571.0 | 17765.0 | 11175.3 |
| RoadRunner | 11.5 | 7845.0 | 39544.0 | 64262.0 | 57900.0 | 50854.7 |
| Robotank | 2.2 | 11.9 | 63.9 | 59.4 | 62.5 | 60.3 |
| Seaquest | 68.4 | 42054.7 | 5860.6 | 8268.0 | 30140.0 | 19652.5 |
| Skiing | -17098.1 | -4336.9 | -13062.3 | -9324.0 | -9289.0 | -9299.3 |
| Solaris | 1236.3 | 12326.7 | 3482.8 | 6740.0 | 8007.0 | 2640.0 |
| SpaceInvaders | 148.0 | 1668.7 | 1692.3 | 20972.0 | 28888.0 | 1749.4 |
| StarGunner | 664.0 | 10250.0 | 54282.0 | 77495.0 | 74677.0 | 60920.6 |
| Tennis | -23.8 | -9.3 | 12.2 | 23.6 | 23.6 | -1.0 |
| TimePilot | 3568.0 | 5229.2 | 4870.0 | 10345.0 | 12236.0 | 6506.4 |
| Tutankham | 11.4 | 167.6 | 68.1 | 297.0 | 293.0 | 231.3 |
| UpNDown | 533.4 | 11693.2 | 9989.9 | 71260.0 | 88148.0 | 36008.1 |
| Venture | 0.0 | 1187.5 | 163.0 | 43.9 | 1318.0 | 993.3 |
| VideoPinball | 16256.9 | 17667.9 | 196760.4 | 705662.0 | 698045.0 | 465578.3 |
| WizardOfWor | 563.5 | 4756.5 | 2704.0 | 25061.0 | 31190.0 | 6132.8 |
| YarsRevenge | 3092.9 | 54576.9 | 18098.9 | 26447.0 | 28379.0 | 2764.4 |
| Zaxxon | 32.5 | 9173.3 | 5363.0 | 13113.0 | 21772.0 | 10806.6 |

Table 4: Raw scores for a single seed across all 55 games, starting with 30 no-op actions. Note that PQR was evaluated on 50M frames. We report the published scores for DQN, QR-DQN, and IQN on 200M frames. Reference values from Yang et al. (2019).

