# OpenReview forum: "Distributional Perturbation for Efficient Exploration in Distributional Reinforcement Learning"
_ICLR.cc/2022/Conference — ICLR 2022 Submitted_

### Official Review · Reviewer_61i6 · 2021-11-03

**Correctness:** 3
**Technical Novelty And Significance:** 3
**Empirical Novelty And Significance:** 3
**Recommendation:** 6
**Confidence:** 3

**Main Review:**

I think that the overall idea is intuitive and makes sense to me. Since committing to specific criterion induces a one-sided tendency on risk for the agent, it seems natural to instead randomize the risk criterion so as to reduce the bias and gradually shrink the ambiguity set to obtain converged results. The paper is to some extend "complete", with both theoretical analysis showing the convergence property as well as a few examples demonstrating the performance of the method.

Additional comments:
1. The writing can be improved. There are many small typos/grammatical errors here and there which seem to indicate that the paper was rushed. Just to name a few: definition 2, line 3, missing "."; P8, "let define a reweigh'; P8, N-Chain, "We more plot the empirical". Please carefully go through the paper and correct the mistakes. Further, the theoretical part contains complicated notation, and the authors could try to see if there is a better way to present the technical content with more intuitive explanations.
2. How were the four Atari games selected? Were they cherry-picked? The authors mention that "To achieve high performance, the agent should take an optimistic approach in some games and a pessimistic approach in others". Perhaps, a more detailed explanation and comparison should be provided so that the readers can better understand the behavior of the method. More thorough experiments and ablations studies on hyper-parameters would greatly improve the paper.

**Summary Of The Paper:**

This paper considers distributional reinforcement learning with the goal of seeking risk-neutral optimal policy. Prior approaches often have a convergence issue due to either being just risk-seeking or risk-averse. Towards resolving the problem, the authors propose a simple, yet effective approach by randomizing the risk criterion during the training process. On the theoretical front, the authors prove that the method converges to an optimal return distribution under certain conditions. Several experimental results also confirm the validity of the method.

**Summary Of The Review:**

Overall, I think the main idea is clearly articulated. While there is room to improve, the theoretical and empirical contributions seem to be relevant and of interest.

---

> ### Author Response · Authors · 2021-11-16
> **Response to Reviewer 61i6**
>
> Thank you for your thoughtful reviews and valuable insights. We appreciate your positive feedback on our paper.
>
> **Q1. Further, the theoretical part contains complicated notation, and the authors could try to see if there is a better way to present the technical content with more intuitive explanations.**
> > We agree with the opinion that there are many complicated notations in the theoretical description. This notation is widely used in the DRO(Distributionally Robust Optimization) literature [Chow et al. 2015, Esfahani and Kuhn 2018, Stanko et al. 2019, Shapiro et al.2021]. We adopt the notations to follow previous DRO works. The notation that we have defined ourselves is only one, $a^*(\xi)$.
> As you suggested, we added the pipeline figure that describes the procedure of our algorithm and the intuitive explanation in the revised version. We also corrected the grammatical and miscellaneous errors in the revised version.
>
> **Q2. How were the four Atari games selected? Were they cherry-picked? The authors mention that "To achieve high performance, the agent should take an optimistic approach in some games and a pessimistic approach in others". Perhaps, a more detailed explanation and comparison should be provided so that the readers can better understand the behavior of the method. More thorough experiments and ablations studies on hyper-parameters would greatly improve the paper.**
> > The results we've shown are not cherry-picked, but because of the very high computation time of Atari environments. Atari games need at least 7 days per seed to learn at 200 million frames. It was impossible to experiment with every 57 games during the submission period, so we selected four Atari environments randomly. LunarLander was an additional experiment to show more results in various environments in that it had a different state configuration compared with Atari. For the rest of the review period, we will add multiple environments with 1 seed in 50M to provide as many experimental results as possible.
> We also agree with your mention that “To achieve high performance, the agent should take an optimistic approach in some games and a pessimistic approach in others” is not well-explained. We removed the sentence in the revised version to avoid unnecessary confusion.
>
> Again, thank you very much for your review and suggestions.
>
> **Reference**
>
> [1] Yinlam Chow, Aviv Tamar, Shie Mannor, and Marco Pavone. Risk-sensitive and robust decision making:
> a cvar optimization approach. arXiv preprint arXiv:1506.02188, 2015.
>
> [2] Peyman Mohajerin Esfahani and Daniel Kuhn. Data-driven distributionally robust optimization
> using the wasserstein metric: Performance guarantees and tractable reformulations. Mathematical
> Programming, 171(1):115–166, 2018.
>
> [3] Silvestr Stanko and Karel Macek. Risk-averse distributional reinforcement learning: A cvar optimization
> approach. In IJCCI, pp. 412–423, 2019.
>
> [4] Shapiro, Alexander, Darinka Dentcheva, and Andrzej Ruszczynski. Lectures on stochastic programming: modeling and theory. Society for Industrial and Applied Mathematics, 2021.

---

### Official Review · Reviewer_Ywdd · 2021-11-04

**Correctness:** 4
**Technical Novelty And Significance:** 2
**Empirical Novelty And Significance:** 2
**Recommendation:** 5
**Confidence:** 4

**Main Review:**

According to the authors, the shortcoming of existing approaches is that they do not separate epistemic and aleatoric uncertainty. This problem appears well motivated (though their example is a bit simplistic; see discussion below). However, I’m not sure I understand how the proposed approach solves the problem. As far as I can tell, their main strategy is to use sampling from a distributional perturbation rather than optimism. The main reason their approach converges appears to be the fact that they require that the size of the perturbation goes to zero over time. Wouldn’t taking the optimism to zero over time achieve the same effect?

Furthermore, their example in Section 3.1 is for a toy algorithm; it is unclear if existing techniques suffer from this issue (though I suspect that they do). As discussed above, it also appears that their algorithm primarily avoids this issue by taking \Delta_n to zero over time; in their example, it appears that you can achieve the same effect by taking c_t to zero over time.

Along the same lines, their theoretical guarantee does not appear to be very strong. For example, as far as I can tell, it applies to traditional Bellman operator (i.e., the one that does not perform any exploration); indeed, this is the case \Delta_n=0 for all n. Thus, their theoretical analysis primarily results from the fact that they are taking \Delta_n to be small sufficiently fast, rather than from any feature of their particular exploration strategy. I would have expected the theory to support the benefits of using their exploration strategy instead of an alternative.

Finally, their experiments also appear to be fairly weak. They only compare to a limited number of baselines; for instance, why not compare to prior approaches such as Keramati et al. 2020? Furthermore, they do not appear to outperform the baselines by very much in most of their empirical results.

Minor comments
- Does Theorem 4 apply for any choice of U_\Delta in Theorem 4? This is not clear.
- The proposed algorithm feels very similar to Thompson sampling; I wonder if some connection can be drawn here.
- The authors do not clearly distinguish that their goal is to use distributional RL, but eventually learn a risk-neural policy. Several of the papers they cite have the goal of learning a risk-aware policy; the concerns regarding epistemic vs. aleatoric uncertainty do not apply to these approaches. It would be helpful if the authors clarify this distinction.


**Summary Of The Paper:**

This paper proposes a novel strategy for performing exploration based on distributional reinforcement learning. They highlight the fact that existing approaches cannot distinguish between epistemic and aleatoric uncertainty. Then, they propose an alternative approach that leverages a random perturbation to the distribution, but then takes this perturbation to zero. They prove that this approach converges. Finally, they validate their approach in their experiments.


**Summary Of The Review:**

Pros
- Important problem

Cons
- Unclear whether their approach solves the stated problem
- Limited theoretical guarantee
- Weak experiments

---

> ### Author Response · Authors · 2021-11-16
> **Response to Reviewer Ywdd (2/2)**
>
> > Furthermore, Theorem 4 and our exploration strategy are closely related to each other, since $\Delta_n$ determines the ambiguity set of distributional perturbation and the distribution perturbation determines the distributional Bellman operator that will be used for selecting the action to be explored.
>
> > Also, Theorem 5, which is described separately, is the main contribution that most other exploration papers [Tang & Agrawal 2018, Mavrin et al. 2019, Clements et al. 2019] have not been able to guarantee. In particular, we want to emphasize that our theoretical work has sufficient meaning in terms of generalization from the existing time-invariant Bellman update to the time-varying case.
>
> **Q4. Why not compare to prior approaches such as [Keramati et al. 2020]? Furthermore, they do not appear to outperform the baselines very much in most of their empirical results.**
> > Since our goal is to explore using risk, we did not compare with [Keramati et al. 2020] because the purposes are different, which aims to create a risk-averse policy. This is because their objective function is not designed to be risk-neutral. On the other hand, in comparison with DLTV, they wanted to apply OFU to risk-neutral problems, so it was an appropriate algorithm as a baseline algorithm. Although we agree that our experimental results do not seem to outperform in terms of performance at 200M frames, it should be noted that the model reached the highest performance faster. This implies that our method is a better exploration method than other baselines. We will add more figures to emphasize our results.
>
> **Q5-1. Does Theorem 4 apply for any choice of $\mathcal{U}_\Delta$ in Theorem 4? This is not clear.**
> > Theorem 4 states that $\mathcal{U}_\Delta$ must not be randomly determined and that the mild condition, the infinite sum of $\Delta$ must be finite, is sufficient to guarantee convergence. If you are asking "any choice under $\mathcal{U}_\Delta$ " rather than "any choice of $\mathcal{U}_\Delta$", then it is possible for any unbiased sampling distribution.
>
> **Q5-2. The proposed algorithm feels very similar to Thompson sampling; I wonder if some connection can be drawn here.**
> > First of all, thank you very much for your insight. The philosophy of Thompson sampling is similar in terms of adopting a randomized strategy. However, to be specific, Thompson sampling is different in that no factor is adjusted for convergence like $\Delta$, and it does not share a common criterion, such as $\xi$. In addition, according to the test results at the beginning of our research, Thompson sampling did not perform well in the presence of aleatoric uncertainty. Hence, we did not provide the result because we did not have enough theoretical background and valid results to support it.
> Although it was not described in this paper due to the paper's page limits, it is attractive enough to study the connection with Thompson sampling in future research.
>
> **Q5-3. The authors do not clearly distinguish that their goal is to use distributional RL, but eventually learn a risk-neutral policy. Several of the papers they cite have the goal of learning a risk-aware policy; the concerns regarding epistemic vs. aleatoric uncertainty do not apply to these approaches. It would be helpful if the authors clarify this distinction.**
> > Citing the papers you mentioned is to emphasize that many papers are used solely to obtain risk awareness policies. As far as we know, there are no studies that use risk in exploration. This is because risk deals with aleatoric uncertainty, whereas exploration aims to reduce epistemic uncertainty. Hence, our works attempt to solve the stuckness of aleatoric uncertainty on exploration and to reduce epistemic uncertainty by only selecting statistically plausible actions.
>
> > Unlike the previous works that learn a risk-aware policy by exploring with the fixed tendency on risk, we randomized the criteria of risk-sensitivity among risk-aversing and risk-seeking exploration by distributional perturbation. In distributional RL, the exploration methods with learned distribution can escape from stuckness easily to achieve better performance in the manner of risk-neutrality. This was mentioned in the introduction and conclusion, but we revised the draft to emphasize it more clearly.
>
> Overall, most of your comments were helpful to improve my paper and thank you for pointing out the way for better follow-up research. However, this is a paper focusing on the first theoretical foundation for an exploration method with the distribution, and the point you are raising is out of the scope of our work. In addition, some of the comments indicate that our key ideas were not well conveyed, so if you have any ambiguities or questions, we will happily answer them and update the draft.

---

> ### Author Response · Authors · 2021-11-16
> **Response to Reviewer Ywdd (1/2)**
>
> Thank you for the detailed comments on our paper.
>
> We identified that OFU exploration in deep RL problems with risk-neutral objectives is problematic because of 'stuckness' which is caused by aleatoric (intrinsic) uncertainty. Specifically, when an agent explores by specific criteria that may have a one-sided tendency on risk, using the estimated uncertainty from the empirical distribution can hinder the convergence and performance. This implies that OFU exploration inherently induces the risk-seeking exploration strategy regardless of the property of objectives. To avoid this issue, we applied a simple but effective approach just by randomizing the risk criterion for exploration in PQR. We further presented an advanced algorithm and theory of convergence. Furthermore, we would like to note that the goal of this paper is to remove a one-sided tendency on risk during exploration under risk-neutral objective (original RL), instead of solving a risk-sensitive (or risk-seeking) reinforcement learning.
>
> One of the cases that take $c_t$ to zero, as you pointed out, is the default setting of DLTV. Despite its risk-neutral purpose, DLTV does not behave as risk-neutral during exploration, since it maintains optimism to aleatoric uncertainty at every step. We have sufficiently shown in Section 3 and 4 that convergence cannot be guaranteed and that the performance is degraded experimentally as shown in Fig 2.
>
> On the other hand, we proposed an efficient exploration method that avoids inferior behaviors specified in [Osband et al. 2019] by selecting only statistically plausible actions that can be greedy with a certain criterion given an ambiguity set. To understand the effect of perturbation intuitively, you can see there are dramatic differences between DLTV and p-DLTV in every experiment.
>
> In addition, we suggested a mild condition that can theoretically guarantee convergence while solving the stuckness caused by aleatoric uncertainty. It is our key contribution that it guarantees convergence for the first time among exploration methodologies in distributional RL.
>
> **Q1. Wouldn't taking the optimism to zero over time achieve the same effect?**
> > We have covered this issue fully in Section 3.1 and Figure 2. p-DLTV performed very effectively compared with DLTV in most experiments, despite it being a simple alternative by replacing optimism with randomization. This means that applying OFU in the distribution RL is not appropriate due to the presence of aleatoric uncertainty which leads to risk-seeking behaviors. Hence, taking the optimism to zero over time does not achieve the same effect at all.
>
> **Q2. As discussed above, it also appears that their algorithm primarily avoids this issue by taking $\Delta_n$ to zero over time; in their example, it appears that you can achieve the same effect by taking $c_t$ to zero over time.**
> > Following the answer for **Q1**, we avoided ‘stuckness’ through randomization. Taking $\Delta_n$ to 0 is to satisfy risk neutrality purpose and to guarantee the convergence. Note that $\Delta_n$ is not a condition for efficient exploration. All experimental results also show that taking $c_t$ to zero (which corresponds to DLTV) does not have the same effect at all as our method.
>
> **Q3. Along the same lines, their theoretical guarantee does not appear to be very strong. For example, as far as I can tell, it applies to the traditional Bellman operator (i.e., the one that does not perform any exploration); indeed, this is the case $\Delta_n=0$ for all $n$. Thus, their theoretical analysis primarily results from the fact that they are taking $\Delta_n$ to be small sufficiently fast, rather than from any feature of their particular exploration strategy. I would have expected the theory to support the benefits of using their exploration strategy instead of an alternative.**
>
>
> > As you mentioned, there is no theoretical description of the benefit, but it has been sufficiently shown by the quantitative and qualitative explanation through Section 3.1 and the experimental results. Since we focus on proposing a novel distributional RL-based exploration method for the first time, we believe that quantitative analysis on the performance of exploration, such as regret in bandit, will be a future work by proposing a suitable metric.
>
> > However, we would like to remark that our theoretical results are valuable. Almost all convergence theorems usually employ some decreasing sequence, such as step size in gradient descent, gamma contraction in value iteration and policy evaluation, and the Robbins-Monro condition for convergence of a stochastic process. Theorem 4 suggests that simply taking $\Delta_n$ to 0 does not guarantee convergence. It tells $\Delta_n$ must satisfy some conditions to guarantee the convergence of the algorithm. Our main contribution is clarifying the condition of $\Delta_n$ to make the algorithm converge. **(continued)**

---

### Official Review · Reviewer_GS3Z · 2021-11-04

**Correctness:** 3
**Technical Novelty And Significance:** 3
**Empirical Novelty And Significance:** 3
**Recommendation:** 6
**Confidence:** 3

**Main Review:**

## Strengths


1. The proposed method is a novel application of results from DRO (Distributionally Robust Optimization) literature to formulate a perturbed bellman optimality operator (PDBOO) that is theoretically well-behaved -- it is shown to converge under a “weak contraction” property and to the same unique fixed point as the standard bellman operator.

2. The pedagogical N-chain environment setting clearly demonstrates the benefit of the proposed risk-neutral method in converging to the optimal action despite the existence of sub-optimal actions that have a low probability of higher reward. In the LunarLander and Atari environments, a similar result is obtained with baselines including DLTV, a perturbed variant of DLTV and QR-DQN.

## Weaknesses

1. The paper is empirically weak, demonstrating results in only 4 Atari environments. It would be good to see a full empirical analysis of all Atari environments. The paper would also benefit from diversity in the choice of environments -- given that it proposes risk-neutral exploration, what about continuous control simulation tasks such as locomotion or manipulation (in MuJoCo for example), or the b-suite family of environments (https://github.com/deepmind/bsuite)?

1. The paper has issues with writing. Grammatical errors are prevalent throughout the paper and the related works, motivation and presentation of the proposed method all seem to be lacking in clarity.



**Summary Of The Paper:**

The paper proposes a novel distributional RL algorithm that is neither risk-seeking (exploratory) or risk-averse (exploitatory) -- but rather risk-neutral. The paper motivates the need for risk neutrality from the family of works applying OFU (optimism in the face of uncertainty) to distributional RL -- it states that prior works in this space induce a one-sided risk tendency (risk-seeking or risk-averse) which is undesirable as it leads to “biased exploration”. An N-chain environment is initially studied where it is shown that prior works (DLTV, QRDQN)  are not able to identify the optimal action as fast as the proposed method (PQR) -- the prior works are misled by the low probability high reward side of the N-chain environment as opposed to the optimal side which provides maximum averaged reward. The paper then proposes a method for perturbing the risk-measure in risk-sensitive RL such that the resulting perturbed distributional bellman operator (PDBOO) converges to the same fixed point as the standard bellman operator (with some weak assumptions) and hence, produces a risk-neutral policy. The paper presented results in the LunarLander-v2 and Atari domain (4 Atari games selected), demonstrating that prior works with one-sided risk tendencies do not perform as well as the proposed risk-neutral method (PQR).

**Summary Of The Review:**

Overall, the paper provides valuable theoretical contributions in the form of their analysis of the perturbed distributional bellman optimality operator (PDBOO) and its successful application in toy environments (N-chain) as well as some non-trivial ones (LunarLander, Atari). However, the paper lacks in quality of writing and empirical analysis.

---

> ### Author Response · Authors · 2021-11-16
> **Response to Reviewer GS3Z**
>
> Thanks for your valuable comments!
>
> **Q1. The paper is empirically weak, demonstrating results in only 4 Atari environments. It would be good to see a full empirical analysis of all Atari environments. The paper would also benefit from diversity in the choice of environments -- given that it proposes risk-neutral exploration, what about continuous control simulation tasks such as locomotion or manipulation (in MuJoCo for example), or the b-suite family of environments?(https://github.com/deepmind/bsuite)**
> > We agree that our experiments did not cover diverse environments. Since each Atari game requires at least 7 days for 200M frames, we tried to choose fast converging environments within 50~100M randomly, instead of running the whole 57 games with 3 seeds. To provide more experimental results, we are running on other Atari for 50M frames with 1 seed to produce additional results of our methods. We will update the results to compare the performances on exploration soon.  Because of the computational issue on Atari, we added LunarLander-v2 which can be learned fast and has a different state configuration compared with Atari. To improve the empirical results of the paper, we are conducting additional experiments on the b-suite environments for the rest of the period. We truly appreciate your environment suggestion.
>
> **Q2. The paper has issues with writing. Grammatical errors are prevalent throughout the paper and the related works, motivation and presentation of the proposed method all seem to be lacking in clarity.**
> >For clarity, we updated our draft overall to emphasize our goal and contribution. Also, we added the pipeline of our algorithm and corrected the mistakes shown in the paper more carefully. We will make an additional section in the appendix for related works in more detail.
>
> Thank you again for the suggestion on how to improve the quality and readability of the paper.

---

### Author Response · Authors · 2021-11-23
**General Response**

We thank all the reviewers for their valuable comments and constructive feedback!

We highlighted the revised sentences in red color for the updated version.

* Clearer explanation and correct the grammatical errors for **all reviewers**.
* Clear notation and proof with better readability for **reviewer 61i6**.
* More reference to improve clarity in the related works for **reviewer GS3Z**.
* Emphasized our novelty and distinguished our goals from risk-sensitive RL for **reviewer Ywdd** throughout the paper.
* Pipeline to understand the progress of our methods for **reviewer GS3Z** and **61i6** in Appendix B.
* More experimental results on full 55 Atari games and comparison with other distributional RL algorithms **for all reviewers** in Appendix D.

If you have any additional concerns or questions, we are willing to discuss more.

Thanks.

---

### Decision · Program_Chairs · 2022-01-20

**Decision:**

Reject

**Comment:**

This paper studies efficient algorithms for distributional reinforcement learning. The motivation stems from the need of risk neutrality, since other existing approaches might have one-sided risk tendencies. The algorithms proposed in this paper are based on sampling from a distributional perturbation rather than using optimism in the face of uncertainty. Both theoretical guarantees and empirical results have been provided to validate the effectiveness of the proposed algorithms. While this is certainly an important and interesting direction, I agree with the reviewers that it is unclear from the theory in this paper why distributional perturbation is helpful.